# Activity-dependent expression of Channelrhodopsin at neuronal synapses

Francesco Gobbo [1,2], Laura Marchetti[1,2,3], Ajesh Jacob[1], Bruno Pinto[1,4], Noemi Binini[5], Federico Pecoraro Bisogni[5], Claudia Alia[1,6], Stefano Luin [2], Matteo Caleo[6], Tommaso Fellin [5], Laura Cancedda [4,7] & Antonino Cattaneo [1]

Increasing evidence points to the importance of dendritic spines in the formation and allocation of memories, and alterations of spine number and physiology are associated to memory and cognitive disorders. Modifications of the activity of subsets of synapses are believed to be crucial for memory establishment. However, the development of a method to directly test this hypothesis, by selectively controlling the activity of potentiated spines, is currently lagging. Here we introduce a hybrid RNA/protein approach to regulate the expression of a light-sensitive membrane channel at activated synapses, enabling selective tagging of potentiated spines following the encoding of a novel context in the hippocampus. This approach can be used to map potentiated synapses in the brain and will make it possible to re-activate the neuron only at previously activated synapses, extending current neuron-tagging technologies in the investigation of memory processes.

[1] Bio@SNS, Scuola Normale Superiore, piazza dei Cavalieri 7, 56126 Pisa, Italy. [2] NEST, Scuola Normale Superiore, piazza San Silvestro 12, 56127 Pisa, Italy. [3] Center for Nanotechnology Innovation @NEST, Istituto Italiano di Tecnologia, piazza San Silvestro 12, 56127 Pisa, Italy. [4] Department of Neuroscience and Brain Technologies, Istituto Italiano di Tecnologia, via Morego 30, 16163 Genoa, Italy. [5] Optical Approaches to Brain Function Laboratory, Department of Neuroscience and Brain Technologies, Istituto Italiano di Tecnologia, via Morego 30, 16163 Genova, Italy. [6] Istituto di Neuroscienze, Consiglio Nazionale delle Ricerche, via Moruzzi 1, 56124 Pisa, Italy. [7] Dulbecco Telethon Institute, via Varese 16b, 00185 Rome, Italy. Correspondence and requests for materials should be addressed to A.C. (email: antonino.cattaneo@sns.it)

U nderstanding the mnemonic processes is one of the greatest challenges in neuroscience. Long-lasting changes in the synaptic connectivity between neurons are generally accepted to be crucial for the establishment and maintenance of memories[1,2]. Similarities between synaptic and memory consolidation suggest shared mechanisms[3–5], and synaptic modifications have been shown to be critically involved in memory formation, strengthening, and recall[6,7]. Recently, it has become possible to define sets of neurons involved in specific memories by activity-dependent tagging[8–10]. However, many details remain to be worked out on the role of the modifications at the synapse level in the encoding and establishment of memories[6,11–13].

Whereas much progress in the understanding of neural circuits has been made using optogenetics[8,9], to date no direct modulation of specific synapses involved in the formation of memories has been possible using state-of-the-art optogenetic tools. Indeed, the current spatial resolution of opsin expression in activity-dependent tagging is the whole neuron. Cell-wide excitation does not take into account, for instance, the complexity of different incoming pathways converging onto the same postsynaptic neuron[14] and the synchronous activation of the whole cell may fail to mimic a physiological condition[14,15]. In recent attempts, subcellular localization of light-sensitive effectors has taken advantage of trafficking signals inserted into the opsin aminoacidic sequence[16–18]. For example, channelrhodopsin-2 (ChR2) and halorhodopsin were differentially targeted with protein-targeting signals to the soma and dendrites of retinal ganglion cells, to recreate antagonistic center-surround receptive fields[19], and fusion with a MyosinVa-binding domain targeted

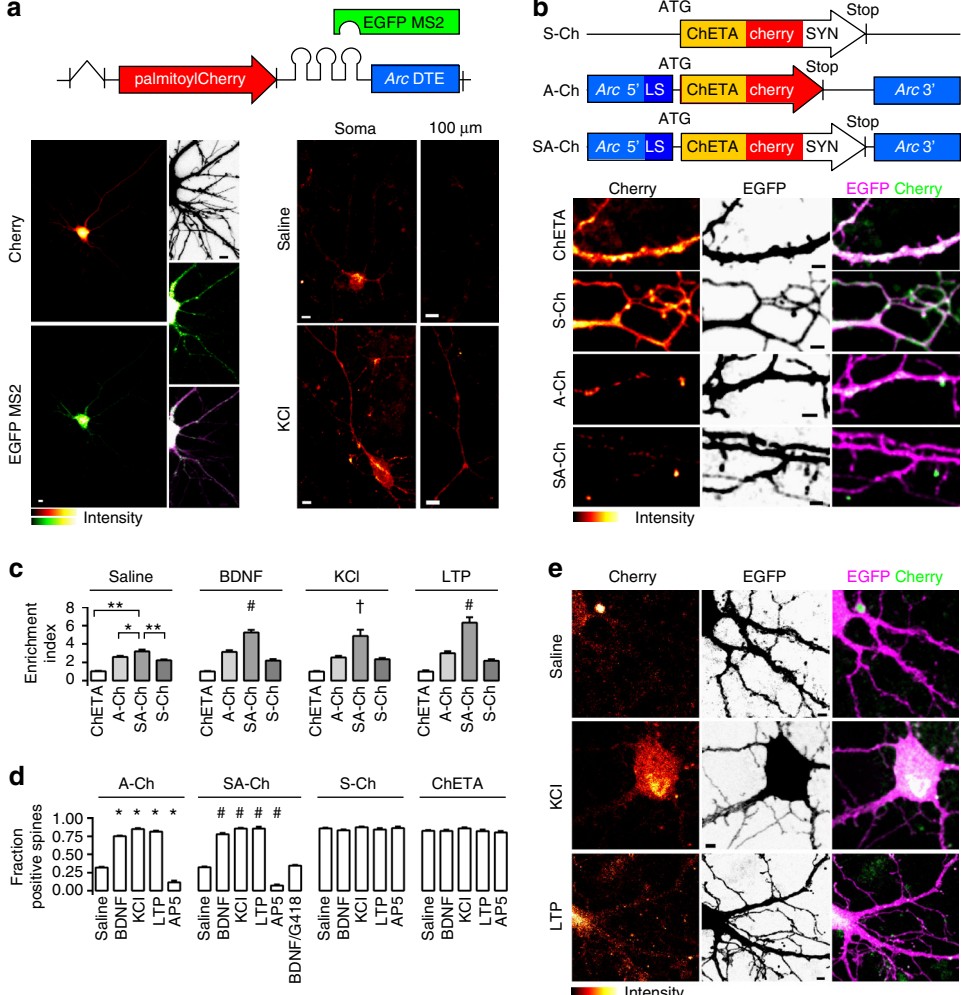

**Fig. 1** Activity-dependent SA-Ch expression at spines. **a** Schematic construct of palmitoyl-Cherry/MS2 reporter. Left, Cherry (top) and EGFP-MS2 (bottom) distribution in living neurons under resting conditions. In the presence of *Arc* DTE, MS2/RNA signal is granular. Inset, top to bottom, neuron profile, EGFP-MS2, merge (stretched levels). Right, *Arc* DTE regulates reporter expression in response to neuron activity. Cherry expression in proximal dendrites ("soma") and 100 μm away from the soma ("100 μm") after 1 h saline (top) or 10 mM KCl (bottom) treatment. **b** Schematic S-Ch, A-Ch and SA-Ch constructs. Below, dendritic pattern of ChETA-Cherry expression (left), EGFP filler (centre) and merge (right) for unmodified ChETA-Cherry and the three constructs above. **c** Enrichment Index for the three constructs and unmodified ChETA-Cherry under different stimulation conditions (see Methods). *$P < 0.01$ and **$P < 0.001$, one-way ANOVA, Bonferroni comparison of means, within group. †$P < 0.05$ and #$P < 0.001$ to SA-Ch, saline treated, one-way ANOVA, Bonferroni comparison of means. **d** Fraction of ChETA-Cherry-expressing spines under different stimulation conditions, grouped for construct. *$P < 0.001$ to A-Ch saline and #$P < 0.001$ to SA-Ch saline, one-way ANOVA, Bonferroni comparison of means. Differences within and between S-Ch and ChETA values are not significant at the 0.05 level. **e** Representative images of SA-Ch-expressing neurons under different treatment conditions. Following KCl or NMDA-dependent LTP, bright ChETA-Cherry puncta are evident along dendrites. Bars are mean ± SEM. Data in **d** and **e** are reported as boxplots in Supplementary Fig. 5. Scale bar (a) 5 μm, (b,e) 2 μm. *N* and replicate numbers for all figures are listed in Supplementary Table 1

ChR2 to the somatodendritic compartment of neurons in living mice[16]. These approaches can be useful to refine spatial stimulation specificity and to activate specific subcellular compartments[18]. However, tagging different subsets of synapses based on their activity in order to selectively stimulate them appear demanding tasks for this protein-based approach, as it is not straightforward to integrate it with activity-tagging methodologies. Indeed, synapses are at the same time a subcellular compartment of the neuron and the physical site of circuit connections, enabling them to undergo local modifications in an autonomous way[6]. Single-synapse optogenetics can be achieved by restricting illumination to single spines[20], but this requires *a priori* knowledge of the identity of the synapses involved in the circuit, in order to test their role in a memory process.

A functionally relevant reactivation of the incoming stimulus in an unbiased, synapse-specific way would require the tagging of activated synapses by locally expressing opsins. Towards this aim, here we describe a novel strategy, named SynActive (SA), for the expression of proteins at synapses in an input-specific, activity-dependent manner by combining RNA targeting elements and a short protein tag. The SA-Channelrhodopsin variant presented here is locally translated at synapses in vitro and in vivo, and the exploration of a novel context increases the number of hippocampal synapses expressing the opsin, revealing a non-random distribution of the activated synapses along dendrites.

## Results

***Arc* mRNA targeting element regulates translation**. We developed a dual RNA/protein reporter to compare possible RNA synaptic tags. Transcripts encode membrane-anchored fast-maturing fluorescent mCherry[21] and bear different dendritic or axonal targeting elements (DTEs and ATEs, see Supplementary Methods); MS2-binding sites in the 3′-untranslated region (UTR) bind EGFP-MS2 protein to visualize RNA[22]. *Arc* is transcribed in an activity-dependent manner and its messenger RNA localizes near synapses that experienced recent activity; in resting conditions, it is believed to be translationally repressed within ribonucleoparticle (RNP) granules[23]. We found that a minimal DTE from *Arc* 3′-UTR[24] determined a significantly lower level of mCherry expression in non-stimulated neurons than strong or constitutive DTEs from alphaCaMKII or MAP2 (Fig. 1a and Supplementary Fig. 1), whereas a discrete, granule-like *Arc*/MS2 signal was detected in the soma and along dendrites (Fig. 1a). KCl activation of neurons expressing the *Arc* DTE construct dramatically increased mCherry fluorescence in dendrites as far as 100 μm away from the soma in as little as 1 h (Fig. 1a), significantly increasing mCherry dendritic pool; conversely, the increase driven by alphaCaMKII DTE was less prominent (Supplementary Fig. 1).

**Synergistic action of RNA and protein**. To enrich opsin expression at synapses, we combined RNA- and protein-targeting sequences. We cloned fast-spiking ChETA-Cherry[25] between *Arc* 5′- and 3′-UTRs. Although 3′-UTR may contain DTEs, 5′-UTR and other parts of 3′-UTR generally regulate translation[26]. For instance *Arc* 5′-UTR has IRES (internal ribosome entry site)-like activity[27], a process involved in the synaptic translation associated to long-term potentiation (LTP)[26]. As ribosomes typically lie at the dendrite-spine junction, we reasoned that a protein tag interacting with postsynaptic components would improve spine retention and enrichment of the newly synthesized protein. We therefore fused to the C terminus of ChETA-Cherry a short bipartite tag (AAAASIESDVAAAAETQV, hereafter SYN tag) composed of the *N*-methyl-D-aspartate receptor (NMDAR) C terminus SIESDV and the PSD95-PDZ-binding consensus

ETQV, which has been previously reported to enrich proteins at postsynaptic sites[28,29].

To compare the distinct contributions of the protein and RNA instructive signals, we generated three constructs and expressed them in primary neurons: (i) *Arc* 5′-ChETA-Cherry-MS2-*Arc* 3′-UTR (hereafter A-Ch); (ii) ChETA-Cherry-SYN tag-MS2 (S-Ch), and (iii) *Arc* 5′-ChETA-Cherry-SYN tag-MS2-*Arc* 3′-UTR (SA-Ch) (Fig. 1b). Neurons expressing the constructs were morphologically similar to each other or to neurons expressing enhanced green fluorescent protein (EGFP) alone; neither the modified SYN-ChETA nor *Arc* UTRs determined significant changes in spine number and morphology (Supplementary Fig. 2a–c). In addition, SA-Ch expression did not alter the ratio of surface NMDAR/AMPAR (Supplementary Fig. 2d).

Brain-derived neurotrophic factor (BDNF) administration, which causes a translation-dependent late form of LTP[30], induced dendritic expression of A-Ch, but not of S-Ch; following BDNF treatment, A-Ch signal in dendrites was significantly higher than that of EGFP, which lacks DTEs and is translated in the soma only (Supplementary Fig. 3a). Conversely, dendritic S-Ch distribution was quite similar to that of EGFP. This is consistent with previous observations that BDNF boosts the translation of transcripts bearing alphaCaMKII 3′-UTR, increasing the protein levels along dendrites as compared to the soma[31]. In addition, A-Ch and SA-Ch RNA in unstimulated neurons, identified by RNA-tethered EGFP-MS2, was prevalently granular along dendrites. Following KCl treatment, the RNA/MS2 signal became much more diffuse (Supplementary Fig. 4a, b), indicating RNA exit from granule, allowing local SA-Ch translation[26].

We then co-expressed the three ChETA-Cherry variants with EGFP in cortical neurons to compare their subcellular expression pattern. S-Ch was enriched at spines compared to unmodified ChETA-Cherry (Fig. 1b, c), but spines were labelled quite evenly. Conversely, A-Ch labelled spines in a sparse way (Fig. 1b). In many cases the base of the spine, rather than the head, was labelled most intensely, and Cherry fluorescence was also prominent on the dendritic shaft. SA-Ch recapitulated the sparse expression pattern typical of A-Ch, while more trustfully tagging spine heads (Fig. 1b).

A quantitative enrichment index (EI), the ratio of ChETA-fused Cherry intensity at the synapse to that measured in the dendritic shaft (1 to 2 μm from the spine junction), demonstrated effective SA-Ch accumulation at synapses. The EI calculated for SA-Ch was significantly higher than that for A-Ch or S-Ch, and all three constructs had higher EI than ChETA-Cherry (Fig. 1c).

**Activity-dependent SA-Ch expression and synapse enrichment**. We next characterized the activity-dependent regulation of SA-Ch expression at synapses. Treatment of cortical neurons with (i) BDNF, that induces L-LTP[26,30], (ii) KCl, and (iii) NMDA, under conditions that promote spine potentiation (NMDA-induced LTP) (see Methods and Supplementary Fig. 6) dramatically increased the number of SA-Ch-positive spines (Fig. 1d, e). Conversely, NMDAR inhibition with AP5 drastically reduced the number of SA-Ch-positive spines. Translation inhibition with G418 (geneticin) blocked BDNF effect on SA-Ch expression, demonstrating its dependence on novel protein synthesis. In terms of expressing spines, SA-Ch response to treatments was identical to that of A-Ch, whereas neither S-Ch nor ChETA-Cherry expression was affected by treatments that increased or decreased neural activity (Fig. 1d). Importantly, treatments that activate neurons or induce synaptic LTP significantly increase SA-Ch EI, relative to saline treatment (Fig. 1c), and A-Ch EI was only modestly responsive to treatments. We ascribe this last effect to the fact that, following translation, A-Ch can diffuse in the

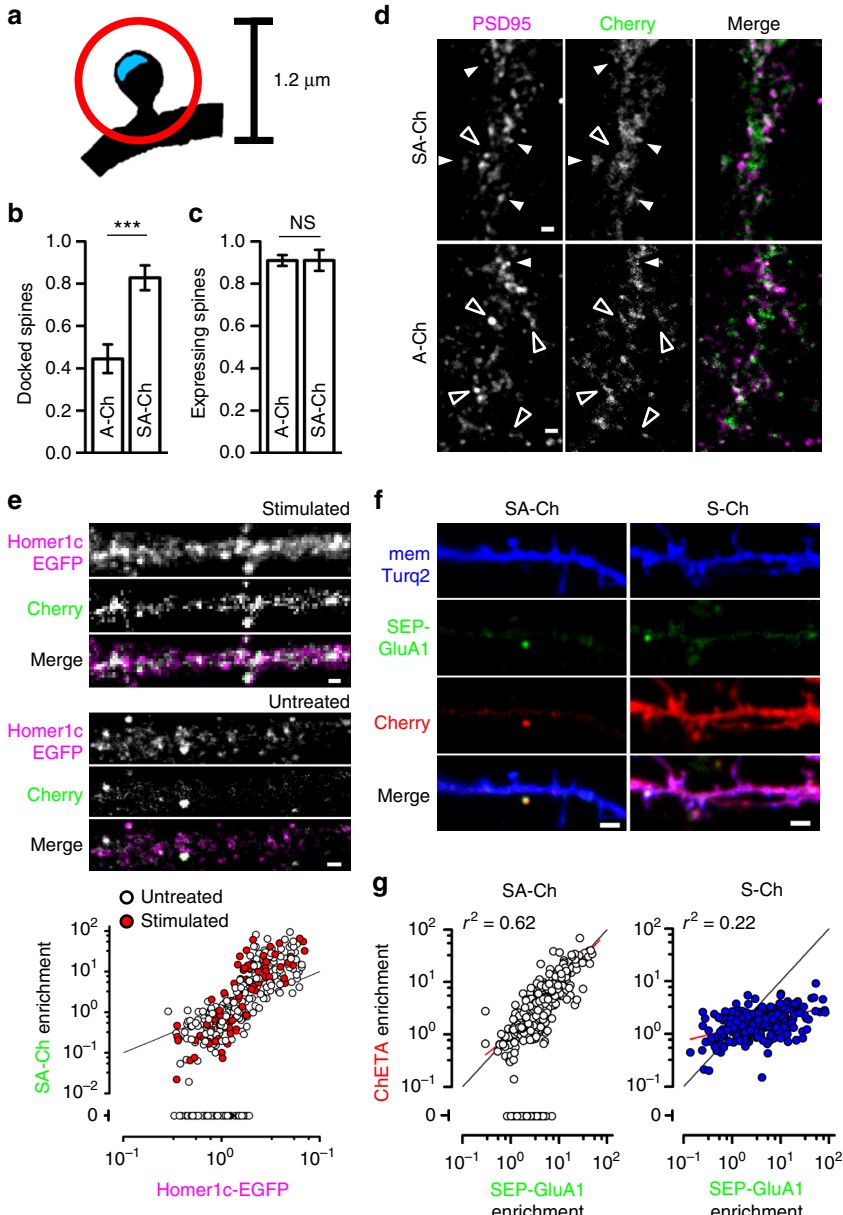

**Fig. 2** Synergistic action of RNA and protein regulatory sequences. SYN tag confers synaptic localization of SA-Ch protein (**a–d**), whereas RNA sequences maps its expression to potentiated spines (**e–g**). **a** Schematic drawing for the determination of "docked" vs. positive but non-"docked" spines. Cherry fluorescence peaks within a circle of 1.2 µm diameter (red circle) centered on the postsynaptic density (PSD - blue area) for positive spines, and on the PSD for "docked" spines. Quantification of "docked" **b** and total positive spines (**c** following cLTP treatment (see Methods) for A-Ch and SA-Ch. Bars are mean ± SEM. ***$P < 0.001$ two-tailed Student's $t$-test. NS, not significant at the $\alpha = 0.05$ level. **d** Representative dendrites of neurons expressing the two constructs. White arrowheads indicate "docked" spines, empty arrowheads positive, non-"docked" spines. Another example is reported in Supplementary Fig. 7a. **e** SA-Ch co-localizes with PSD marker Homer1c-EGFP. In stimulated as well as in unstimulated neurons, SA-Ch was expressed at Homer1c-EGFP puncta (top). SA-Ch correlation with Homer1c-EGFP is supralinear, indicating that SA-Ch is preferentially enriched at spines with larger PSD (bottom graph). Spines that do not express SA-Ch are assigned a value of zero. White dots are spines from unstimulated neurons, red dots from stimulated ones, black line represents the diagonal. Because the plot has double-log scale, any linear correlation has unitary slope and is parallel to this line. **f** Comparison of SA-Ch and S-Ch expression: SA-Ch is only expressed at SEP-GluA1-tagged spines, whereas S-Ch has no preference for SEP-GluA1-positive spines. **g** SA-Ch (grey dots) significantly correlates with SEP-GluA1 expression in a linear fashion, whereas S-Ch (blue dots) does not. A proportion of SEP-GluA1-positive spines do not express SA-Ch and those spines are assigned an enrichment value of zero. Red lines indicate the regression lines. Scale bar, 1 µm (**d**, **e**) and 2 µm (**f**)

membrane both onto the spine head and along the dendritic shaft; conversely, the SYN tag helps retention of SA-Ch in the spine (Fig. 1c). The observed somatic SA-Ch protein (Fig. 1e and Supplementary Fig. 4) can probably be ascribed to the global level of the stimulations, which can signal the overexpressed transcript to be de-repressed also in the soma. In fact, in non-stimulated

neurons somatic expression is much lower (Fig. 1e, first row and Supplementary Fig. 4a, second row), and can be further reduced by controlling promoter strength and localizing stimulation (see section "In vivo synaptic tagging with SA-Ch").

To probe the specificity of SA-Ch accumulation at synapses, we performed double immunofluorescence (IF) against Cherry and

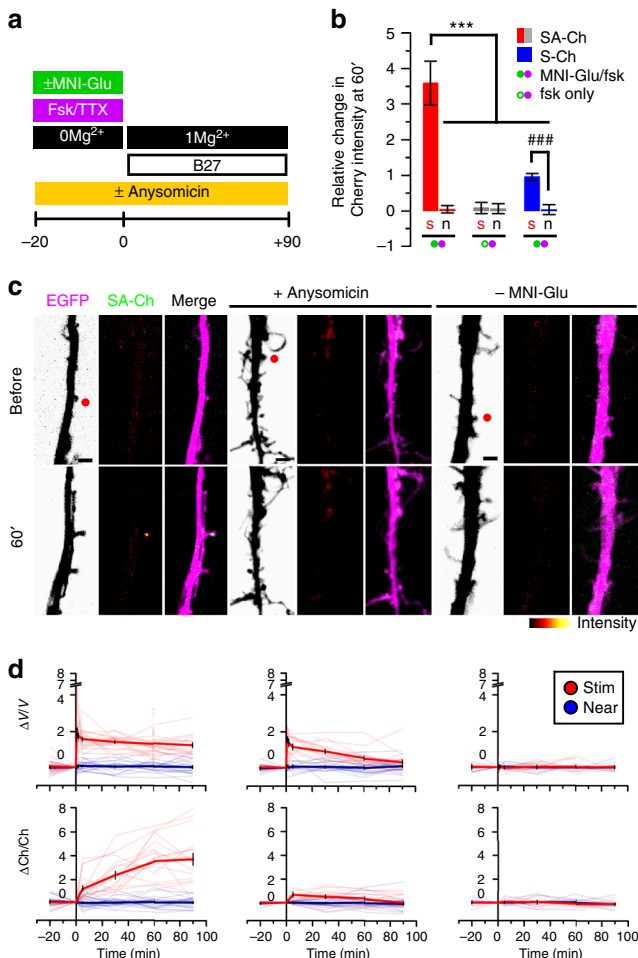

**Fig. 3** Synapse specificity of SA-Ch expression at potentiated synapses. SA-Ch is specifically expressed at potentiated synapses. DIV 8-10 neurons were focally stimulated by uncaging glutamate in close proximity to selected spines. **a** Neurons were maintained in standard $Mg^{2+}$-free ACSF in the presence of forskolin and TTX for 20 min before uncaging with or without MNI-caged glutamate. Following two-photon uncaging, medium was changed to 1 mM $Mg^{2+}$ ACSF supplemented with B27. **b** Local release of glutamate stimulates SA-Ch translation at stimulated (s), but not nearby (n) spines. This effect was specific to glutamate release, as it was absent when MNI-glutamate was not added to the medium. Following stimulation, S-Ch change was much lower and is an effect of spine enlargement. ***$P < 0.001$, one-way ANOVA, Bonferroni comparison of means. ### $P < 0.001$ unpaired samples Student's $t$-test, two-tailed. Bars are mean ± SEM. **c** Translation inhibition with anysomicin blocked SA-Ch accumulation at stimulated synapses. Representative images of stimulated dendrites in neurons transfected with SA-Ch. Red dots in the EGFP channel indicate the location of two-photon uncaging. Experimental conditions are indicated on top of images. Scale bar, 2 μm. **d** Time course of relative changes in volume ($\Delta V/V$, top graphs, measured by the EGFP intensity) and SA-Ch intensity ($\Delta Ch/Ch$, bottom graphs) following uncaging of stimulated (red) and near spines (blue). Stimulation induced a long-lasting volume change, paralleled by a slowly rising accumulation of SA-Ch; in the presence of anysomicin, volume change was transient and no accumulation of SA-Ch was evident. Bold lines represent mean ± EM, whereas narrow lines are single traces for depicted data for stimulated (light red) and nonstimulated (light blue) spines. **d** Corresponding conditions in **c** above: from left to right, samples with MNI-Glu/forskolin, MNI-Glu/forskolin/ anysomicin, forskolin only/no MNI-Glu. Open circles are corresponding $\Delta V/V$ values at 60 min for SA-Ch spines in **b**

PSD95, a major component of the post-synaptic density (PSD), and compared the localization of A-Ch and SA-Ch, following NMDAR-dependent LTP. Spines were considered "docked" if mCherry signal coincided with that of PSD95, and positive, but not "docked", if it peaked outside the PSD, but within a circle of 0.6 μm radius centered on the PSD (Fig. 2a). About half of the spines expressing A-Ch were "docked" ($44 \pm 7\%$), whereas these constituted the vast majority ($83 \pm 6\%$) of SA-Ch spines (Fig. 2b). The total number of positive spines was the same for both constructs (Fig. 2c). Representative images are presented in Fig. 2d and in Supplementary Fig. 7a.

Above data demonstrate the strong dependence of SA-Ch expression on neural activity. This suggests that, in untreated cultures, positive spines received a sustained stimulation from the spontaneous activity of the culture, which could amplify basal NMDAR activation[32]. Consistently, blocking NMDAR activity with AP5 drastically reduces A-Ch and SA-Ch expression (Fig. 1d), and SA-Ch localized preferentially to larger PSDs, as indicated by the supralinear correlation between SA-Ch enrichment at synapses and Homer1c-EGFP content, an excellent indicator of spine volume and PSD size[33,34] (Fig. 2e). As spine enlargement strongly correlates with functional potentiation[35], this suggests SA-Ch expression at potentiated synapses; indeed, a significant proportion of spines with lower Homer1c content were devoid of any SA-Ch signal in non stimulated neurons. In addition, SA-Ch was preferentially expressed at synapses exposing the AMPAR-subunit 1 fused to supereclyptic-EGFP (SEP-GluA1), a marker of functional potentiation[36] (SA-Ch-positive spines were $63 \pm 3\%$ of SEP-GluA1-positive spines, and $6 \pm 2\%$ of SEP-GluA1-negative spines, $P < 0.001$, $\chi^2$-test). The relationship between SA-Ch and SEP-GluA1 enrichment was linear (Fig. 2f), indicating that SA-Ch is expressed at potentiated synapses. Notably, some of the spines with lower SEP-GluA1 enrichment did not express SA-Ch, most likely because they received a weaker stimulation. In fact, AMPAR exocytosis takes place during E-LTP phase, which has a lower threshold than translation-dependent L-LTP[3,36]. On the contrary, S-Ch enrichment only showed a modest dependence on SEP-GluA1 (Fig. 2g).

Experiments in hippocampal neuron cultures yield almost identical results to cortical neurons: SA-Ch co-localized with Homer1c-EGFP and NMDA-LTP strongly increased the number of SA-Ch-positive synapses (Supplementary Fig. 7b, c). Altogether, these data demonstrate the activity dependence of SA-Ch translation, as well as its preferential localization at postsynaptic sites.

**Synapse-specificity of SA-Ch expression.** We demonstrated synapse specificity of SA-Ch expression by focally stimulating selected synapses from neurons expressing EGFP and SA-Ch with two-photon glutamate uncaging in the presence of the protein kinase A(PKA) activator forskolin[37]. Tetrodotoxin (TTX) prevented potentiation from spontaneous activity under these elevated cAMP conditions, as well as synaptic capture. Glutamate uncaging induced SA-Ch expression at stimulated, but not at neighbor synapses or at other synapses on the same dendrite (Fig. 3). When caged glutamate was absent, no significant change in SA-Ch intensity was observed following focal illumination (Fig. 3b). Stimulated spines showed a sustained increase in volume (Fig. 3d), a structural rearrangement that parallels functional potentiation[35]. The increased Cherry intensity at stimulated spines observed for S-Ch, which is translated exclusively in the soma (Supplementary Fig. 3), is likely due to the PSD expansion following the volume change, and was significantly lower than what observed for SA-Ch (Fig. 3b).

It is unlikely that the increase in SA-Ch at the potentiated synapse is due to protein mobilization from surrounding regions, as no significant change in intensity in neighboring spines and in the dendritic shaft was apparent. The time course of SA-Ch increase also rules out this possibility, because it shows a slow rising phase following the stimulation that reaches a plateau between 30 and 60 min (Fig. 3d). Most importantly, translation inhibition with anysomicin blocked SA-Ch accumulation at stimulated spines (Fig. 3c, d) and the change in spine volume observed after stimulation slowly declined to pre-stimulation levels (Fig. 3d). Thus, synapse potentiation drives local SA-Ch expression in a protein-synthesis-dependent, synapse-specific way.

**Optogenetic activation of SA-Ch-tagged synapses.** Having established synapse specificity of SA-Ch expression, we asked whether the locally synthesized SA-Ch is effective in driving synaptic currents. Calcium influx is an established indicator of spine activation, both in vitro and in vivo[38–40]. We therefore co-expressed SA-Ch with the green calcium indicator GCaMP6s[38]. To minimize ChETA activation while imaging GCaMP6s, we stimulated SA-Ch spines with laser scanning at 488 nm wavelength and imaged GCaMP6s stimulating at 990 nm with two-photon excitation[41,42] in a region encompassing the base of the spine and the corresponding dendrite. We observed light-induced $\Delta F/F$ calcium transients in most stimulated spines, but not when blue light stimulation was omitted; consistently, TTX inhibition of presynaptic activity did not influence the recording of light-induced calcium transients (Fig. 4a). Channelrhodopsins are weakly permeable to calcium[43], but their stimulation could lead to the opening of voltage-gated calcium channels (VGCCs). Accordingly, blue light stimulation performed in the presence of VGCC inhibitors nifedipine, $Ni^{2+}$, and $Zn^{2+}$ induced a markedly reduced response (Fig. 4a). In neurons expressing untargeted ChETA-Cherry, both the spine and the nearby dendrite could evoke light-dependent $\Delta F/F$ calcium transients when illuminated, in accordance with ref. [20]; on the contrary, only spines, and not dendrites, of SA-Ch-expressing neurons were responsive to blue light stimulation (Fig. 4b). SA-Ch expression does not appear to alter the normal synaptic transmission, as spontaneous $\Delta F/F$ calcium events that could be sometimes recorded from SA-Ch spines were not significantly different from those recorded from control neurons expressing palmitoyl-Cherry and GCaMP6s (Supplementary Fig. 8).

To see whether the activation of Channelrhodopsin-tagged synapses would mirror a physiological activation based on neurotransmitter release, we expressed SA-Ch or unmodified ChETA-Cherry in hippocampal neurons and determined CaMKII phosphorylation by IF 7.5 min after optogenetically stimulating them with a light pattern similar to $\theta$ burst stimuli used to induce LTP in the hippocampus (see Methods). In fact, sustained glutamate release activates synaptic CaMKII and determines its rapid phosphorylation that lasts for minutes[44]. To reduce background CaMKII phosphorylation, spontaneous activity was pharmacologically suppressed with TTX and glutamate receptors inhibitors, for the 3 h preceding light stimulation (Fig. 4c). During and after illumination, action potentials were inhibited with TTX. Light stimulation strongly increased phospho-CaMKII signal in SA-Ch-expressing neurons compared with neurons that were maintained in the dark. Light alone had no effect, as EGFP-only-expressing neurons were not affected by the stimulation and synaptic levels of phospho-CaMKII were comparable to those in unstimulated neurons expressing SA-Ch (Fig. 4c, d). Importantly, in optically stimulated neurons CaMKII phosphorylation was specific to SA-Ch-positive spines, as spines from the same neuron

lacking Cherry signal did not differ from non-stimulated neurons. Indeed, physiologically, CaMKII activation is specific to stimulated spines[44]. Cell-wide activation of untargeted ChETA-Cherry also activated CaMKII, although the synaptic phospho-staining was lower than for SA-Ch (Fig. 4c). Interestingly, phospho-CaMKII staining was evident in the dendritic shaft of illuminated ChETA-Cherry, but not of SA-Ch neurons (Fig. 4d). A possible explanation is that, in ChETA-Cherry-expressing neurons, part of the CaMKII pool fails to translocate from the shaft into the spine due to concomitant extrasynaptic depolarization, as also data presented in Fig. 4b suggest; conversely, localized SA-Ch activation could more readily induce CamKII phosphorylation and mobilization, just as neurotransmitter-mediated synapse stimulation mobilizes CaMKII from the dendritic shaft and accumulates it at the spine head[44]. We conclude that large-field optical stimulation of synaptic SA-Ch is able to simulate an input-specific excitation onto the postsynaptic neuron; conversely, whole-cell activation of ChETA-Cherry has a different outcome on the neuron response at the subcellular level.

We next asked whether the optical activation of SA-Ch synapses could also drive global neuronal activation by illuminating cultured hippocampal neurons with blue light pulses as above and evaluating c-fos expression, an immediate early gene that is induced in neurons shortly after strong synaptic stimulation[45]. Light-stimulated neurons expressing SA-Ch and EGFP displayed evident nuclear c-fos staining 1 h after optogenetic activation (Supplementary Fig. 9). Conversely, c-fos staining was lower in control cells transfected with SA-Ch and EGFP that were maintained in the dark. Exposure to light alone had no effect on c-fos expression, as illuminated neurons that expressed EGFP only had lower levels of nuclear c-fos, comparable to those in SA-Ch, not stimulated neurons (Supplementary Fig. 9). Thus, optogenetic stimulation of neuronal cultures demonstrates that optical activation of SA-Ch-expressing synapses by large-field illumination is able to recapitulate key features of neuron-to-neuron communication.

***In vivo* synaptic tagging with SA-Ch.** Our work in culture demonstrates that SA-Ch is expressed at potentiated spines in cortical and hippocampal neurons. We next sought to investigate SA-Ch behavior in vivo, in order to (i) compare its somatic vs. synaptic expression and (ii) evaluate its expression following a natural stimulus such as the exploration of an unfamiliar environment, a paradigm that rapidly activates *c-fos*, as well as *Arc* expression in the hippocampus of mice and rats[45,46]. Blockage of *Arc* translation or of general protein synthesis with *Arc* antisense oligo-nucleotides or anysomicin inhibits context memory[3,47]. Indeed, a large body of evidence has identified populations of cells that are activated in the hippocampus when animals are presented a novel context[8,45]. We expressed SA-Ch under the Tet-responsive TRE promoter[48] in the hippocampus by means of triple-electrode *in utero* electroporation[49]. Mouse embryos were co-electroporated with constitutive transactivator rtTA and Tet-responsive EGFP. SA-Ch and EGFP transcription is induced with intraperitoneal administration of the tetracycline analog doxycycline, allowing us to restrict synapse tagging by SA-Ch to a defined time window. Control mice that did not receive doxycycline showed no expression. SA-Ch expression in the hippocampus in vivo showed a remarkable synaptic selectivity, as it was detected almost exclusively at spines of electroporated neurons, whereas somas were largely devoid of Cherry fluorescence (Fig. 5a and Supplementary Fig. 10). Conversely, untargeted ChR2 labeled intensely both dendrites in the stratum radiatum and stratum oriens, and somas in the stratum pyramidale (Supplementary Fig. 10c).

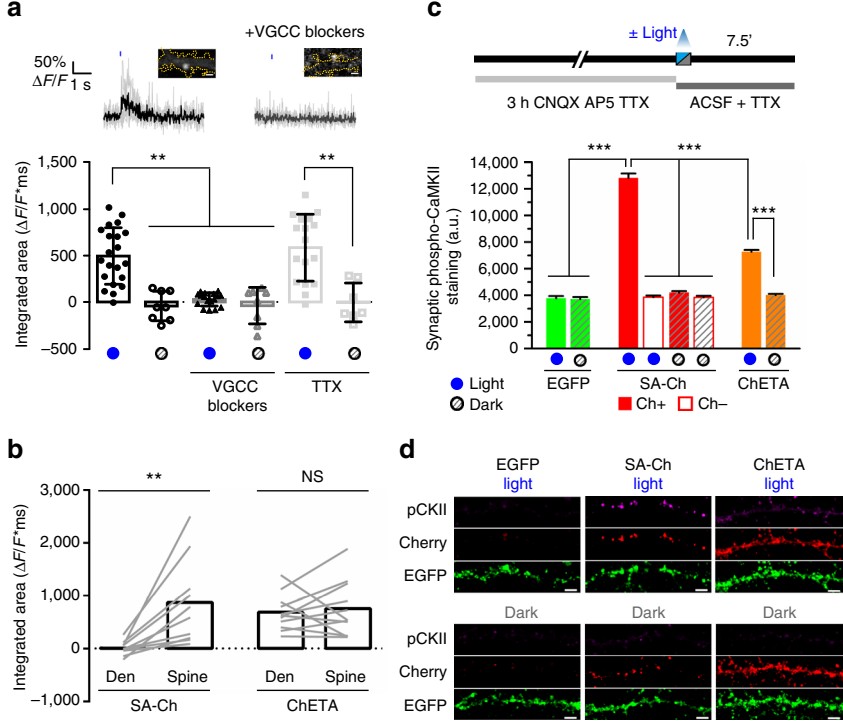

**Fig. 4** Optogenetic activation of SA-Ch in neurons. **a** Activation of SA-Ch by means of 488 nm laser illumination drives calcium influx in the neuron. GCaMP6s was expressed along with SA-Ch and illuminated at 990 nm. Top, example of GCaMP6s $\Delta F/F$ time course following 10 ms light stimulation (blue mark above) in ACSF (left, black trace) or in ACSF with voltage-gated calcium channel (VGCC) blockers nifedipine, $Ni^{2+}$, and $Zn^{2+}$ (right, gray trace). Single traces are in light gray and bold lines are the average. Above images show SA-Ch fluorescence in corresponding spines and yellow dotted trace shows neuron profile as inferred from GCaMP6s fluorescence; scale bar, 1 μm. Bottom, integrated area of $\Delta F/F$ against time plot after light stimulation for imaged spines with (blue dots below) or without (gray dots) light stimulation. SA-Ch neurons were recorded in standard ACSF (black, circles) or in ACSF in the presence of either VGCC inhibitors (dark gray, triangles) or TTX (light gray, squares); open symbols represent recordings in the same medium as corresponding filled symbols, without illumination. Data are values for single spines, each represents the average of trains on the same spine, bars are mean ± SD. **P < 0.01, Kruskal–Wallis test, Dunn's *post-hoc* comparisons. **b** Illumination of spines in SA-Ch-expressing neurons drives calcium influx, but not when laser illumination is focused on the nearby dendrite. Conversely, ChETA neurons can be excited both by a spine-focused and a dendrite-focused laser beam. **P < 0.01 paired Student's *t*-test, two-tailed. NS, not significant at the $\alpha = 0.05$ level. Lines connect single paired data points, bars are mean. **c** Outline of time course of the experiment. Cells were pretreated for 3 h with CNQX, AP5, and TTX, to reduce background CaMKII activation. Neurons were fixed 7.5 min after light stimulation and stained for phospho-CaMKII. Spines in SA-Ch-expressing neurons were subgrouped into Cherry-positive (filled bars) and Cherry-negative spines (empty bars). Light stimulation induced a significant increase of phospho-CaMKII staining in SA-Ch-expressing spines ***P < 0.001, one-way ANOVA, Bonferroni comparison of means. Bars are mean ± SEM. **d** Representative images of data shown in **c**. Panels show phospho-CaMKII immunofluorescence (p-CKII), anti-Cherry immunofluorescence (Cherry), and EGFP signal. Scale bar, 5 μm

To address possible concerns that targeting SA-Ch at the synapse may alter synaptic physiology, we electroporated mice as above, with TRE:SA-Ch, constitutive rtTA, and soluble mCherry, and we induced SA-Ch expression with doxycycline. We recorded AMPA- and NMDA-EPSCs in CA1 pyramidal neurons, while stimulating the Schaffer collateral with a bipolar electrode, and calculated the NMDA/AMPA ratio[50]. Importantly, the NMDA/AMPA ratio was not significantly different in CA1 neurons expressing the transgene (Cherry-positive cells) and in control non-electroporated CA1 neurons (Cherry-negative cells, Supplementary Fig. 11).

Next, electroporated mice with TRE:SA-Ch and TRE:EGFP were treated for 3 days with doxycycline and on the fourth day, while still on doxycycline, were exposed to a novel context with unfamiliar visual cues; control animals remained in the home cage. The exposure to a novel context significantly increased the number of SA-Ch spines in the CA1 and in the dentate gyrus (DG) regions (Fig. 5b, c). Interestingly, both in CA1 and in the DG, SA-Ch spines were closer to each other than what we calculated by randomly shuffling their positions along dendrites

(Fig. 5d); consistently, the closest non-expressing spine was located at a greater distance than what would be expected by chance (Supplementary Fig. 12a). Not only the first neighbor spine, but also the second and third neighbor spines were significantly more likely to express SA-Ch than what would happen by chance (Fig. 5e). This implies the existence of clusters of SA-Ch-expressing spines and, hence, of potentiated synapses. We defined a cluster of potentiated synapses as a set of SA-Ch synapses comprising at least two spines separated by no more than 2 μm, i.e., two spines belong to the same cluster if their interdistance is <2 μm. In home-caged animals, 86% and 85% of spines in CA1 and DG, respectively, belonged to a cluster and the exposure to a novel context increased this proportion to 95% and 94%, respectively. As clusters ranged from 2 to 13 spines, context exploration significantly increased the average dimension of spine units, i.e. clusters and single spines taken together, both in the CA1 and in the DG (Fig. 5f).

SA-Ch spines also appeared to be non-randomly distributed across different dendrites: some dendrites had a higher density of potentiated synapses than nearby dendrites. This was more

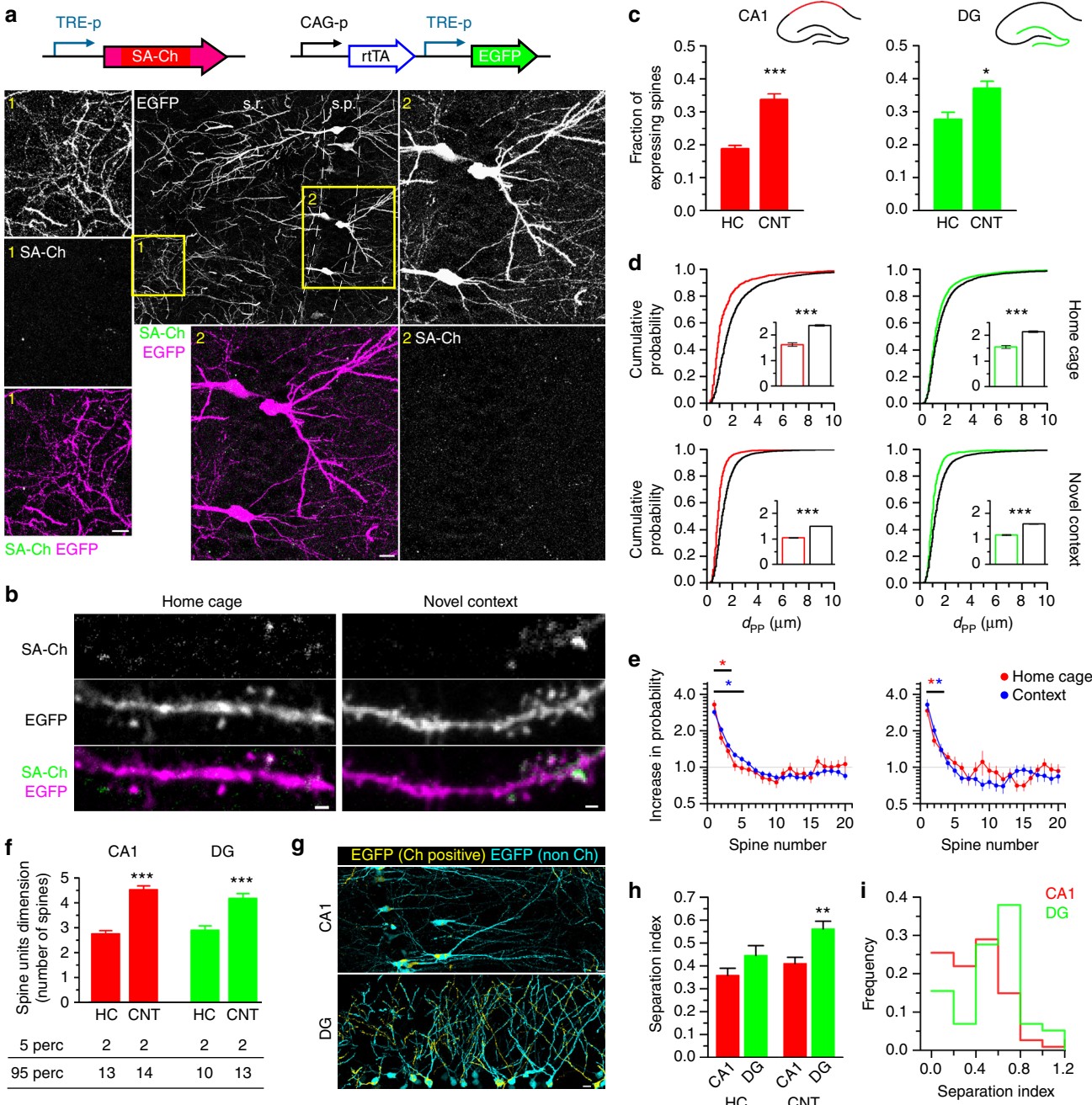

**Fig. 5** *In vivo* synaptic tagging. **a** Constructs expressed in the hippocampus of electroporated mice. TRE-p is the tetracycline-responsive promoter. Expression of SA-Ch in mouse CA1 after 2d intraperitoneal doxycycline injection. Dendritic (1) and somatic (2) regions (yellow squares in the large-field image) are magnified, showing EGFP, SA-Ch, and merge channels (SA-Ch: green, EGFP: magenta). s.p. stratum pyramidale, s.r. stratum radiatum. **b** Dendrites expressing SA-Ch in pyramidal neurons in CA1 from animals held in the home cage or exposed to a novel context (see text). **c** Average fractions of SA-Ch-positive spines in CA1 and DG neurons in home cage (HC) and novel context (CNT) groups. *$P < 0.05$, ***$P < 0.001$, Student's *t*-test, two-tailed. **d** Distribution of first neighbor distance ($d_{PP}$) for two potentiated spines in CA1 (left) and in the DG (right). Graphs on the top represent data from home-caged animals, bottom graphs from animals exposed to a novel context. Black lines are the corresponding distributions after randomly shuffling positive/negative assignments to the original spine positions. Insets show, with corresponding colors, first neighbor distance mean ± SEM. ***$P < 0.001$ Kruskal–Wallis test, followed by Dunn's comparison. **e** Measured increase in estimated probability to be potentiated themselves for the first 20 neighbors of a potentiated spines in CA1 (left panel) and DG (right panel) neurons. Under the assumption of random distribution of potentiated spines, this value should be 1 uniformly. *$P < 0.01$, *z*-test from the reference value of 1. Points are mean ± SEM. **f** Mean number of spines per unit (cluster and single spines). Bottom table represents the 5th and 95th percentiles of cluster dimension. ***$P < 0.001$ Kruskal–Wallis test, Dunn's comparison. Bars are mean ± SEM. **g** Localization of SA-Ch-positive regions (yellow) and SA-Ch-negative region (cyan). Original images are in Supplementary Fig. 12. **h** Separation index (see text for details). A large separation index value indicates that the number of potentiated spine in the dendrite strongly deviates from the expected value if potentiated spines were equally distributed in neurons from the same slice. **$P < 0.01$ Kruskal–Wallis test, Dunn's comparison. **i** Separation index distribution of dendrites in CA1 and the DG in context-exposed animals. Scale bar, 10 μm (**a**, **g**) and 1 μm (**b**).

strikingly observed in the DG granule cells, whereas CA1 pyramidal neurons were more uniform (Fig. 5g and Supplementary Fig. 12b). We calculated the difference between the number of SA-Ch spines in each dendrite and the number that would be expected if potentiated spines were uniformly distributed in dendrites in the same region of the slice (Separation Index, see Methods). We found that in the new context-exposed mice the dendrites of DG granule cells had a significantly higher Separation Index (SI) than CA1 pyramidal neurons (Fig. 5h). In addition, the majority of DG neurons had a high SI, implying that, in the DG, neurons were divided in two populations with either a high or a low number of potentiated synapses (Fig. 5i).

## Discussion

The current paradigm in memory studies, based on theoretical and experimental data, points to synaptic ensembles underlying the generation and storage of new memories[6,12,13]. Accordingly, learning different tasks involves different sets of spines, which further supports the idea that spines, and not whole neurons, are the more relevant entities for information storage in the brain[51,52]. At the same time, methods for activity mapping and causally probing memories heavily rely on promoters from immediate early genes[9,13].

We propose here the SA approach to refine the investigation on memory (and other) circuits to map brain activity at the synaptic scale. pSA plasmids combine activity-dependent translation at potentiated synapses with *Arc* mRNA regulatory sequences and a protein tag to drive synaptic expression of any desired gene (including, but not limited to, opsins).

Our findings are consistent with most observations implying a role for *Arc* in synapse potentiation[23,53]. SA-Ch is preferentially found at larger spines (Fig. 2e) and is expressed at focally stimulated spines (Fig. 3). We tested regulatory sequences and DTEs from other mRNAs, including BDNF, which is also targeted to dendrites in an activity-dependent manner[54]. However, the BDNF splice variants that we employed (exons IIa, IIc, and VI), out of the many BDNF transcripts[54], were much less responsive to neural activity, either due to high basal translation (IIa and IIc) or to almost undetectable translation competence (VI form). The *Arc* sequences that we tested gave the best results; however, we envisage that background expression can be further lowered to increase synaptic enrichment; in any case, the use of less strong or tunable promoters (as the TRE promoter) can also be helpful, as our data in vivo suggest. This would also help further reducing the residual somatic expression.

Current imaging techniques to label synapses, such as mGRASP, can be modified to restrict synapse mapping to determinate regions, projections or cellular types with long-established genetic or tracing technologies[55]. On the other hand, efforts in the implementation of activity sensors have made it possible to record real-time synapse activity in response to sensory stimulations[40,56]. However, activity alone does not imply the involvement in the storage of a defined status and not all active synapses become potentiated[39,56]. Recently, activity reporter SEP-GluA1, which labels synapses incorporating fluorescent AMPA receptor subunit 1 on the membrane surface, has been proposed as a marker for synapse potentiation[36,57]. AMPA receptors are rapidly exposed on the surface of spines that undergo sustained stimulation[57], which is generally accepted to be responsible for the increased currents following potentiation. However, potentiation comprises dissociable events, and different forms of potentiation exist[58]. Some do not last indefinitely and AMPA receptors incorporation may be transient[33]. Instead, our strategy can act as reporter of a late-phase, translation-dependent LTP (L-LTP)[58] and can be used to map potentiated synapses across a

population of neurons in memory tasks, thus enabling to identify candidate "synaptic engrams." Indeed, SA-Ch significantly correlates with SEP-GluA1 accumulation on postsynaptic sites; however, spines with lower SEP-GluA1 enrichment were also in many cases devoid of SA-Ch (Fig. 2f), suggesting that SA proteins would tag the subpopulation of SEP-GluA1-expressing spines that undergo L-LTP, a likely candidate for memory storage unit in the brain[13].

Work in acute hippocampal slices identified potentiated spines in CA1 with the incorporation of fluorescent phalloidin[59]. Recently, the incorporation of fluorescently tagged AMPA receptors has been observed in vivo in the mouse barrel cortex following whisker stimulation[60], providing the first observation of potentiated spines in vivo. Here we demonstrate the usefulness of the SA approach by labeling synapses that underwent translational-dependent potentiation in the hippocampus of live mice exploring a novel context. Exposure to a novel environment has been linked to an increase in active neurons, as identified by c-fos staining or catFISH technique[46,47], but no parallel has been done so far with long-lasting synaptic changes such as translation-dependent LTP.

Previous work identified functional clusters of synapses in cortical areas and hippocampal slices (spines with correlated activity)[36,40,61]; clustering has been proposed in models for cooperative integration of synaptic activity in neuron computation, sensory integration, and memory formation[62–64]. We identified clusters of potentiated synapses in hippocampal regions CA1 and DG, which comprised most of the potentiated spines. It must be noted that the exact number of spines in each cluster is dependent on the chosen cutoff (2 μm) in our working definition. However, we consider this choice reasonable, when taking into account the physical dimensions of a typical mushroom spine. Indeed, the range (2–13 spines) that we calculated for the cluster dimension (Fig. 5f) is in accordance to what reported for functional clusters identified by calcium imaging of synaptic activity (2–12 synapses)[61].

Our approach enabled us to map potentiated synapses across different dendrites, highlighting differences in the distribution between the DG and the CA1 regions (Fig. 5h, i). Our data suggest that single dendrites of granule cells function as a highly homogeneous unit in terms of activity integration and plasticity, supporting a role of the DG for pattern separation[15,65]. According to this model, DG cells encode highly orthogonal contextual information, whereas downstream CA3 and CA1 complete and process this information. Consistently, whole-cell optogenetic activation of engram cells in the DG, but not in CA1, served as an effective contextual stimulus in the fear conditioning protocol[9], despite other experiments clearly advocating a role for CA1 in the encoding of contextual information[66].

In order to highlight the role of potentiated synapses in a memory recall framework, it is necessary to envisage an experimental strategy to selectively act on them, but cell-wide neuron activation also recruits other learning-related mechanisms at the cellular level[67]. Although subcellular optogenetic stimulation can be achieved by restricting the illumination pattern down to single spines[20], this requires *a priori* knowledge of the sites to be stimulated, which are not always known. Moreover, the feasible number and sparseness of distinct illumination spots heavily depend on technological aspects. On the other hand, the biologically achieved spatial restriction of Channelrhodopsin expression presented here, would allow unbiased excitation of recently activated synapses with standard experimental setups for wide-field illumination. In this scenario, light power should be adjusted so that the effect of the optical stimulation is similar to physiological synaptic events; from our results in culture, we have found that although blue light reactivation of the locally expressed SA-

Ch is able to elicit calcium transients in a specific manner, these evoked calcium signals look somewhat smaller than calcium events occurring in the same spines spontaneously (Supplementary Fig. 9). However, it must be noted that ChETA itself does not have the greatest photocurrents among the opsin family[25,68] and a number of ChR2 variants now exists with larger photocurrents[69]. The majority of these variants differ from the parental ChR2 by a few point mutations[68], so we expect that the substitution of ChETA complementary DNA to encode an opsin with a higher photoconductance would replicate the key expression features of SA-Ch in tagging potentiated synapses. Changing the fluorescent proteins attached to SA-Ch could also increase photocurrents, as Cherry-fused ChR2 have been sometimes reported to have a reduced trafficking to the plasma membrane than fusion proteins of the GFP family[70].

SA-Ch application (or any of its relatives) could help clarifying the role of synaptic potentiation in the formation and recall of encoded memories. Synapse re-excitation could be performed more physiologically than what existing technologies used to tag and reactivate whole neurons can achieve. For instance, the work presented in this paper lays the ground for the use of SA-Ch to test the hypothesis of a "synaptic engram," parallel to the identified "population engram"[6,12,13]. It is likely to be that the two activity-tagging approaches (cellular vs. synaptic) would give similar results where there is large identity overlap between the unit of plasticity and the single neuron, as in the DG[15,63,71]. On the other hand, CA1 neurons receive multiple converging inputs whose crosstalk, following activation by current whole-cell optogenetic protocols, is likely to result in memory occlusion[9].

Taking advantage of *Arc* RNA regulatory sequences, we were able to express a Channelrhodopsin variant at synapses undergoing potentiation, establishing a novel tool to map and reactivate these sites. Recently, a novel approach towards the development of "synaptic optogenetic" strategies was proposed[72]; by expressing a photoactivable form of Rac1 in the motor cortex, Kasai and colleagues[72] demonstrated that the light-induced shrinkage of recently potentiated spines severely impairs motor learning. That study emphasizes the necessity of controlling selected inputs, rather than a selected population of neurons, underscoring the interest of synaptic optogenetic approaches, such as the one presented here. However, by dramatically altering actin dynamics, such approach determined a drastic alteration of the spine structure; therefore, the interference with the memory trace could not be reverted. Accordingly, it was not possible to perform a memory recall task, as the intervention was purely destructive. Although establishing a first important step in highlighting, and interfering with, established engrams at the synaptic level, the sufficiency of those potentiated synaptic inputs for memory encoding remains to be addressed. Our approach allows, in principle, to re-excite those synapses. In addition, it is likely to be naturally extended to any variant opsin family, thus enabling the bidirectional interference of the synaptic inputs involved in circuit traces and memories.

## Methods

**Constructs**. Palmitoyl-Cherry-MS2 was generated by cloning palmitoylation sequence MLCCMRRTKQ from GAP43 to Cherry N-terminal, whereas MS2 sequence was derived from plasmid pSL-MS2 12X (Addgene 27119). *Arc* DTE comprises nucleotides 2035–2701 of *Arc* transcript (NCBI NM_019361.1), in accordance to ref. [24]. EGFP-MS2 coat protein-NLS was constructed and cloned into pcDNA3.1(+) (Invitrogen) from plasmid Cherry-MS2 coat protein-NLS (a gift from A. Marcello, ICGEB Trieste). ChETA-Cherry cDNA was PCR amplified from plasmid pAAV-CaMKII-hChR2 (E123A)-mCherry-WPRE[25]. ChETA-Cherry-SYN (S-Ch) was generated by cloning 5′-GCCGCCGCTGCTTCAATTGAAAGT-GACGTGGCCGCAGCTGCCGAAACCCAGGTGTAATAA-3′ oligo sequence (IDT Technologies) in frame to ChETA-Cherry using unique site BglII site at 3′-end of Cherry cDNA. A-Ch and SA-Ch constructs were generated by inserting *Arc*

5′- and 3′-UTRs before and after ChETA-Cherry and S-Ch cDNA, respectively. *Arc* UTRs were amplified from plasmid pCMV-ArcF encompassing whole 5′-UTR and first 13 nucleotides of *Arc* CDS, where start ATG was mutated to ACG, and whole 3′-UTR[24]. MS2 sequence was inserted downstream STOP codon before 3′-UTR. Constructs were cloned into plasmid pcDNA3.1(+) (Invitrogen) under cytomegalovirus (CMV) promoter. EGFP was expressed from plasmid pN1-EGFP (Clontech). Homer1c-EGFP was kindly provided by D. Choquet, Institut interdisciplinaire de Neurosciences CNRS, Université Bordeaux 2. Palmitoyl-Turquoise2 is Addgene plasmid 36209. GCaMP6s was expressed from pGP-CMV-GCaMP6s (Addgene 40753). SEP-GluA1 was expressed from Addgene plasmid 64942. For *in utero* electroporation, SA-Ch was inserted downstream of third-generation TRE promoter[48] in a plasmid containing the minimal CK0.4 promoter driving the expression of rtTA2S-M2 transactivator amplified from vector TMPrtTA[73], yielding pTRE3-SA-CK-rtTA. Parental plasmid was custom synthesised by Life Technologies (USA). It was cotransfected with plasmid pCAGGS-rtTA-TRE-EGFP, which was generated by cloning rtTA2S-M2 and TRE-EGFP sequences into plasmid pCAGGS[49]. TRE-EGFP was amplified by PCR from plasmid pSIN-TRE-EGFP, provided by Dr L. Marchetti. pCAGGS-rtTA-IRES-mCherry was generated analogously and IRES sequence was derived from pCAGGS[49].

**Cell culture**. Primary cortical and hippocampal neurons were extracted from P0 B6129 mice as follows: after surgery and tissue isolation, tissue was triturated in cold calcium-free Hank's balanced salt solution with 100 U ml$^{-1}$ penicillin, 0.1 mg ml$^{-1}$ streptomycin, and digested in 0.1% trypsin, followed by inactivation in 10% fetal bovine serum (FBS) Dulbecco's modified Eagle's medium (Invitrogen) 100 U ml$^{-1}$ DNase. Neurons were seeded on previously poly-D-lysine-coated glass coverslips or plasma-treated poly-D-lysine-coated Willco dishes. For initial plating, neurons were maintained in Neurobasal-A medium (Invitrogen) supplemented with 4.5 g l$^{-1}$ D-glucose, 10% FBS, 2% B27 (Invitrogen), 1% Glutamax (Invitrogen), 1 mM pyruvate, 4 μM reduced glutathione, and 12.5 μM glutamate. From the following day on, neurons were grown in Neurobasal-A medium (Invitrogen) supplemented with 2% B27 (Invitrogen), 1% Glutamax (Invitrogen), and 1–10 μg ml$^{-1}$ gentamicin. Medium was refreshed every 2–4 days. For experiments in Fig. 1a and Supplementary Fig. 1, div 12 neurons were used. All other experiments employed div 17–19 neurons. Neurons were transfected with calcium phosphate method the day before experiment. The procedure was approved by the National Council for Research Ethical Committee.

**Treatments**. Neurons as in Fig. 1 were treated for 1 h with either KCl to a final concentration of 10 mM, or with saline, added to bath. Otherwise, treatments are (i) BDNF: hBDNF (Alomone) 100 ng ml$^{-1}$ 90′; (ii) KCl: KCl 10 mM 90′; (iii) LTP: 20′ in 2 mM CaCl$_2$, 1 mM MgCl$_2$ ACSF (artificial cerebrospinal fluid: 136 mM NaCl, 2.5 mM KCl, 10 mM glucose, 2 mM sodium pyruvate, 1 mM ascorbic acid, 0.5 mM myo-inositol, 10 mM HEPES pH 7.3, with 2 mM CaCl$_2$ and 1 mM MgCl$_2$ unless otherwise indicated) followed by 10′ in 2 mM CaCl$_2$/Mg$^{2+}$-free ACSF, 5.4 mM KCl, 100 μM NMDA (Sigma-Aldrich, Saint Louis, MO), 20 μM glycine (Sigma-Aldrich), and 0.1 μM rolipram (Sigma-Aldrich) as described[74], followed by 90′ in culture medium; (iv) AP5: 50 μM AP5 (Sigma-Aldrich) from transfection to analysis (17–20 h). See also Supplementary Fig. 6 for temporal outline of treatments. Stimulated neurons in Fig. 2e are treated with 20′ 2 mM CaCl2, 1 mM MgCl$_2$ ACSF followed by 5′ in 2 mM CaCl$_2$/Mg$^{2+}$-free ACSF, 60 mM KCl, 100 μM NMDA (Sigma-Aldrich), 20 μM glycine (Sigma-Aldrich), and fixed after 90′ (see below).

**Immunofluorescence**. Neurons expressing A-Ch or SA-Ch were fixed in 2% formaldehyde 5% sucrose phosphate-buffered saline (PBS) and permeabilized in 0.1% Triton X-100. After PBS washing, samples were blocked in 1% bovine serum albumin (BSA) PBS, and primary antibodies anti-Cherry (GeneTex GTX59788, 1:500) and anti-PSD95 (Abcam ab9909, 1:600) were used in 0.5% BSA PBS. After washing, primary antibodies were detected with anti-rabbit-TRITC (Sigma-Aldrich T6778, 1:200) and anti-mouse-Alexa647 (Thermo Fisher A32728, 1:200) in 0.5% BSA PBS. Coverslips were mounted in Fluoroshield (Sigma-Aldrich) mounting medium. Hippocampal neurons expressing EGFP, ChETA/EGFP, S-Ch/EGFP, or SA-Ch/EGFP for 24 h were processed as above. Primary antibody was 1:2,500 anti-MAP2 (Abcam ab5392) and it was detected with anti-chicken-Alexa647 (Abcam ab150171, 1:250). For surface NMDAR/AMPAR immunostaining, div 9 neurons were transfected with SA-Ch and palmitoyl-Turquoise2, or palmitoyl-Turquoise2 alone; on the third day from transfection, neurons were fixed in 4% formaldehyde, 5% sucrose PBS and washed, blocked in 5% BSA PBS, and stained with 1:500 anti-GluR1-NT (Millipore MAB2263) and 1:500 anti-GluN1 (Alomone AGC-001), and followed by 1:200 anti-mouse-Alexa488 (Thermo Fisher A32723)/1:200 anti-rabbit-Alexa647 (Thermo Fisher A32733) and mounting.

**Microscopy**. Optical sections (512 × 512 pixels) were acquired with a confocal microscope (Leica TCS SP5 SMD on an inverted DM6000 microscope) using an oil objective HCX PL APO CS 40 × (numerical aperture NA = 1.25), and pinhole was set to 1.47 AU. Digital zoom was adjusted for sampling spines correctly. For whole-cell reconstruction, z-stacks were acquired every 0.5 μm. Sequential illumination

with HeNe 633, Ar 561, Ar 488, Ar 458, and diode (Picoquant, Berlin, Germany) 405 laser lines was used for Alexa647, TRITC and Cherry, EGFP and Alexa488, Turquoise2 and 4′,6-diamidino-2-phenylindole (DAPI), respectively. Neurons in Fig. 1a and Supplementary Fig. 1 were acquired with the same acquisition parameters.

For two-photon uncaging, images were acquired using an Olympus FV1000 confocal module on an inverted IX81 microscope with immersion oil objective UPLSAPO 60× (NA = 1.35) and pinhole was set to 180 μm. Digital zoom was set to 8x. Used laser lines were Ar 488 and HeNe 543 for EGFP and Cherry excitation, respectively. For two-photon uncaging, 720 nm line was set on a tunable Chameleon Vision II Ti:Sapphire pulsed laser (Coherent, 80 MHz). Green and red channels were acquired before 720 nm stimulation (−5′ time point) and 60′ after medium change (see the following two-photon uncaging section).

**Two-photon uncaging.** DIV 8–10 cortical neurons were seeded on plasma-treated, poly-D-lysine-coated Willco dishes, and transfected the day before experiment. Neurons were maintained in $Mg^{2+}$-free ACSF (in mM, 136 NaCl, 2.5 KCl, 2 $CaCl_2$ 10 D-glucose, 10 HEPES, 2 pyruvate, 1 ascorbic acid, 0.5 myo-inositol) with 10 μM forskolin (Tocris BioSciences), 1 μM TTX (Tocris BioSciences) and, where indicated, 2.5 mM MNI-caged glutamate (Tocris BioSciences, Bristol, UK) for 20′ before uncaging. Following EGFP and Cherry acquisition, 30 pulses (720 nm, 9–13 mW at the objective lens) of 7 ms were delivered at 0.5 Hz at 0.5–1 μm from spine head as in ref. [37]. After 5′, medium was changed to 1 mM $MgCl_2$ ACSF supplemented with 2% B27 and the same dendrite was imaged after 60′. The mock stimulation was conducted in the same way, except that MNI-glutamate was not added in the medium. For time-course experiments, red and green channels were acquired 20′ and 5′ before the uncaging start. Green channel was acquired at 0.5′, 1′, 2′, 5′, 30′, 60′, and 90′ following uncaging, and red channel was acquired at 5′, 30′, 60′, and 90′. Throughout the whole protocols, neurons were maintained at 37 °C under humidified 5% $CO_2$ atmosphere. In experiments with translation inhibitors, 5 μM anysomicin (Sigma) was present in the medium all the time staring from the 20′ preincubation.

**Calcium imaging.** Div 7-11 cortical neurons grown on glass-bottom coverslip, expressing GCaMP6s and SA-Ch were imaged using an Olympus FV1000 confocal module on an inverted IX81 microscope with immersion oil objective UPLSAPO 60× (NA = 1.35). Neurons were maintained in ACSF containing 2 mM $CaCl_2$, 1 mM $MgCl_2$ at 37 °C under humidified atmosphere. In a parallel sets of experiments, we included (i) VGCC inhibitors nifedipine 5 μM (Sigma), $Ni^{2+}$ (as $NiSO_4$) 500 μM, and $Zn^{2+}$ (as $ZnCl_2$) 500 μM, or (ii) 1 μM TTX (Tocris Biosciences). SA-Ch was imaged at 543 nm and GCaMP6s was excited with Chameleon Vision II Ti: Sapphire pulsed laser (Coherent, 80 MHz) tuned at 990 nm (actual peak was detected at 988 ± 2 nm), to minimize Channelrhodopsin excitation[42]. Selected spines were identified comparing the 543 nm and the two-photon channel. A rectangular imaging region of interest (ROI) was defined on the dendrite immediately under the selected spine, whereas the excitation ROI was set on the spine. We acquired 500 frames every 20 ms by exciting at 990 nm using RM690 filter; GCaMP6s fluorescence was acquired in the 500–600 nm range. The size and dimension of the ROIs were maintained constant in all experiments. After 50 frames, we stimulated the spine with a 10 ms pulse of the 488 nm laser line in spiral scanning mode in the excitation ROI and continued imaging. Laser power (488 nm) was measured to be 8.9–10.7 μW upon steady illumination[41] and 990 nm laser power was 2.5–3.7 mW. Randomly between stimulations, trains were performed identically, except the 488 nm laser line was kept switched off. After dark frame subtraction, $\Delta F/F$ values were integrated for the first 200 frames following stimulation.

For the recording of low-frequency spontaneous events, two-photon GCaMP6s imaging of a region under a defined spine was performed as above, except the duration of each recording session was extended up to 40 s. Neurons transfected with palmitoyl-Cherry-MS2 or SA-Ch, and GCaMP6s were recorded in ACSF; spontaneous events from SA-Ch were derived from long recording sessions without stimulation of SA-Ch expressing spines that were responsive to light (see above).

Neurons in Fig. 4b expressing SA-Ch or ChETA-Cherry and GCaMP6s were recorded as described above. The recording ROI under the spine was maintained constant across recording sessions, whereas the excitation ROI was set on the spine or on the dendrite adjacent to the imaging ROI; for each session, both recordings with spine-centered and dendrite-centered excitation ROIs were performed. The position of the excitation ROI for the first recording of a session (i.e., spine or dendrite) was chosen randomly.

**Culture optogenetics.** DIV 17–19 hippocampal neurons were grown on poly-D-lysine-coated glass coverslips in 24 wells. The day after transfection, neurons expressing SA-Ch and EGFP, ChETA-Cherry and EGFP, or EGFP alone, were put in standard 2 mM $CaCl_2$ 1 mM $MgCl_2$ ACSF and illuminated with single-channel PlexBright LED Module 450 nm connected to an optical fiber (THORLABS, 200 μM diameter, 0.39 NA, ceramic ferrule) at 1–3 mW peak power (measured at the end of the fiber). Ten trains of 13 pulses at 100 Hz were repeated at 0.5 Hz; 4 stimulations at different positions were performed on each culture, in order to

evenly illuminate the whole culture area. In a first set of experiments, neurons were pre-treated for 3 h with 40 μM CNQX, 100 μM AP5, and 1 μM TTX. Medium was changed to ACSF, 2 mM $CaCl_2$, 1 mM $MgCl_2$, and 1 μM TTX, and cultures were light stimulated or maintained in the dark; 7.5 min after stimulation, neurons were fixed for 15 min in 2% formaldehyde, 5% sucrose PBS supplemented with 1 mM $Na_2VO_4$ (Sigma-Aldrich), and 1 mM NaF (Sigma-Aldrich) to inhibit phosphatases; after permeabilization in ice-cold methanol, neurons were blocked in 5% BSA, 1 mM $Na_2VO_4$, 1 mM NaF PBS, and subsequently incubated overnight with 1:100 mouse anti-phosphoCaMKII (Thermo Fisher MA1-047 clone 22B1) and 1:300 rabbit anti-Cherry (GeneTex GTX59788) in 2% BSA PBS. Secondary antibodies were 1:100 anti-rabbit-TRITC (Sigma-Aldrich T6778) and 1:100 anti-mouse-Alexa647 (Thermo Fisher A32728) in 2% BSA PBS. In the second set of experiments, neurons were light-stimulated in ACSF, 2 mM $CaCl_2$, and 1 mM $MgCl_2$; after stimulation, neurons were put back into culture medium; parallel cultures did not undergo such a treatment and were maintained in the dark. After 1 h, cells were fixed in 2% formaldehyde, 5% sucrose PBS, and permeabilized in 0.5% Triton X-100; after PBS washing, cells were blocked in 4% BSA PBS and hybridized with 1:100 rabbit polyclonal anti c-fos (Santa Cruz sc-52) in 2% BSA and 0.05% Triton X-100 PBS. Secondary antibody was 1:100 anti-rabbit-Alexa647 (Thermo Fisher A32733). Samples were mounted in Fluorishield with DAPI (Sigma-Aldrich).

***In utero*** **electroporation and animal experiments.** Animal care and experimental procedures were approved by the IIT and CIBIO licensing, as well as with the Italian Ministry of Health. Hippocampal *in utero* electroporation was performed as described in ref. [49]: E15.5 timed-pregnant CD1 mice (Charles River SRL, Italy) were used. Time-pregnant matings were performed on the evening; the day after mating was defined as E0.5 and the day of birth was defined as P0. The plasmids pTRE3-SA-CK-rtTA and pCAGGS-rtTA-TRE-EGFP (or, alternatively, pTRE3-SA-CK-rtTA and pCAGGS-rtTA-IRES-mCherry) were used at 1 μg μl$^{-1}$ with Fast Green dye (0.3 mg ml$^{-1}$; Sigma) to allow visualization. Briefly, the dam was anesthetized with isofluorane (induction, 4.0%; surgery, 2.0%) and the uterine horns were exposed by laparotomy. Each embryo was injected (3–4 μl) through the uterine wall unilaterally with a 30 G needle (Pic indolor, Grandate, Italy). For the electroporation, 6 electrical pulses (amplitude, 30 V; duration, 50 ms; intervals, 1) were delivered with a square-wave electroporation generator (CUY21EDIT; Nepa Gene). Then, the uterine horns were returned into the abdominal cavity and embryos were allowed to continue their normal development. Mice from both sexes were P24–P26 on the day of the experiment. A first group of mice received 0.5 mg doxycycline (1 mg per 30 g BW) in saline solution intraperitoneally once a day for 2 days; on the third day, brains were fixed by transcardial perfusion of 4% formaldehyde. A second group of mice received an additional intraperitoneal injection (1 mg per mouse) on day 3 and brains were fixed on day 4. On day 4, mice of the novel context group were put separately in a different cage (novel context). Two of the walls had visual cues (3 cm black/white vertical stripes and 3 × 3 cm black/white dashboard); one object was put in the cage (a blue 50 ml Falcon tube)[57]. After 3 h, brains were fixed by 4% formaldehyde transcardial perfusion. Home-cage animals received the same doses of doxyxycline and were kept in their cage until perfusion. After perfusion, brains were post-fixed overnight in 4% formaldehyde in PBS, then cryoprotected in 30% sucrose PBS. Sixty micrometers of coronal sections were cut with a cryostat. Slices were mounted in Vectashield or Fluoroshield with DAPI (Sigma) and native fluorescence was imaged with Leica SP5 (see above) with 1.5 AU pinhole; stacks encompassing the whole section were acquired every 0.5 μm. To calculate intensity profiles in CA1 large fields (Supplementary Fig. 10), IF was performed on free floating slices. Slices were blocked 1 h in 0.3% Triton X-100 in 10% normal goat serum (NGS, Sigma-Aldrich) PBS, then incubated overnight in 1:500 anti-GFP (Abcam ab38689), 1:500 anti-Cherry (Abcam ab16743), in 0.3% Triton X-100, and 10% NGS PBS at 4 °C, washed three times (10′ each), and incubated in secondary antibodies (1:100 anti-mouse-Alexa488 (Thermo Fisher A32723) and 1:100 anti-rabbit-Alexa647 (Thermo Fisher A32733)), 1:200 in 0.3% Triton X-100, and 10% NGS for 1 h. After three washes in PBS, slices were mounted in Fluoroshield with DAPI (Sigma-Aldrich).

Brains from Thy1-ChR2-YFP (line 18) mice (B6.Cg-Tg(Thy1-COP4/EYFP) 9Gfng/J, Jackson Laboratory, stock 007612) strongly expressing ChR2 in CA1 and only modestly in CA3 and DG, were fixed with 4% formaldehyde transcardial perfusion. Brains were post-fixed overnight in 4% formaldehyde in PBS, then cryoprotected in 30% sucrose PBS. Coronal sections of 60 μm were cut with a cryostat and mounted in Fluoroshield with DAPI.

**Slice electrophysiology and analysis.** Acute coronal slices from the mouse neocortex were prepared at postnatal day 21–25. Slices were cut from animals previously electroporated at E15.5 with pCAGGS-rtTA-IRES-Cherry and pTRE3-SA-CK-rtTA (for details regarding *in utero* electroporation, see previous section) and kept for 4 days on doxycycline (intraperitoneal injection, 0.5 mg per day). Mice were anesthetized with urethane (1.65 g kg$^{-1}$) and the brain was quickly dissected and placed in an ice-cold cutting solution containing: 130 mM K-gluconate, 15 mM KCl, 0.2 mM EGTA, 20 mM HEPES, and 25 mM glucose, pH adjusted to 7.4 with NaOH. The solution was constantly oxygenated. Slices (thickness: 300 μm) were first cut with a vibratome (VT1000S, Leica Microsystems, GmbH, Wetzlar, Germany) and immersed for 1 min in solution at room temperature (RT) containing: 225 mM D-mannitol, 25 mM glucose, 2.5 mM KCl, 1.25 mM $NaH_2PO_4$, 26 mM

$NaHCO_3$, 0.8 mM $CaCl_2$, 8 mM $MgCl_2$, pH 7.4 with 95% $O_2$/5% $CO_2$. Slices were then incubated for 30 min at 35 °C in sACSF composed of: 125 mM NaCl, 2.5 mM KCl, 25 mM $NaHCO_3$, 1.25 mM $NaH_2PO_4$, 2 mM $MgCl_2$, 1 mM $CaCl_2$, 25 mM glucose, pH 7.4 with 95% $O_2$/ 5% $CO_2$. After incubations slices were maintained in sACSF at RT until use. During experiments, slices were positioned in a submerged recording chamber (RC-26, Warner Instruments, Hamden, CT, USA) and continuously perfused with fresh bathing solution (125 mM NaCl, 2.5 mM KCl, 25 mM $NaHCO_3$, 1.25 mM $NaH_2PO_4$, 2 mM $MgCl_2$, 2 mM $CaCl_2$, 25 mM glucose, pH 7.4 with 95 %$O_2$/5 % $CO_2$) including the $GABA_A$-receptor antagonist picrotoxin (0.1 mM, Sigma-Aldrich) and maintained at 30–32 °C by an inline solution heater (TC-344B, Warner Instruments). Pipettes (resistance: 3-4 MΩ) were filled with intracellular solution containing: 8 mM NaCl, 145 mM Cs-methanesulfonate, 10 mM HEPES, 10 mM phosphocreatine di(tris) salt, 2 mM $Na_2ATP$, 0.5 mM NaGTP, 0.3 mM EGTA, 5 mM lidocaine N-ethyl bromide, and 10 mM tetraethylammonium chloride, pH adjusted to 7.25 with CsOH. Only recordings with series resistance < 20 MΩ were included in the analysis. Series resistance was not compensated and data were not corrected for the liquid junction potential. Electrical signals were amplified by a Multiclamp 700B, low-pass filtered at 2 kHz, digitized at 50 kHz with a Digidata 1440 and acquired with pClamp 10 (Molecular Device, Sunnyvale, CA). Electrophysiological traces were analyzed using Clampfit 10.4 software (Molecular Device).

Synaptic responses were recorded in the same animal in whole-cell voltage-clamp configuration from Cherry-positive CA1 pyramidal cells of the electroporated hemisphere and from Cherry-negative CA1 pyramidal cells of the non electroporated hemisphere. To evoke synaptic responses, Schaffer collaterals were stimulated at 0.1 Hz (stimulus duration: 100 µs) with a bipolar metal electrode placed in the CA1 stratum radiatum, about 100–200 µm from the recording site. Stimulus intensity was adjusted to obtain half-maximal AMPA receptor-mediated excitatory postsynaptic currents (AMPA-EPSCs, average stimulating current: 129 ± 34 µA). AMPA-EPSCs were recorded at $V_m = -80$ mV and completely blocked in bathing solution containing 2,3-Dioxo-6-nitro-1,2,3,4-tetrahydrobenzo[f] quinoxaline-7-sulfonamide disodium salt (NBQX, 10 µM, Tocris Bioscience). NMDA receptor-mediated EPSCs (NMDA-EPSCs) were recorded at $V_m = +40$ mV in the presence of 10 µM NBQX. The NMDA/AMPA ratio was computed as the ratio of the NMDA-EPSC peak amplitude to the AMPA-EPSC peak amplitude (both measured on a mean trace obtained averaging ten consecutive traces) in the same cell.

**Data quantification**. Spine number and subclass for neurons in Supplementary Fig. 3 were assigned manually based on established nomenclature. Short spines with no apparent neck are classified as stubby; elongated spines whose head and neck diameters are similar are classified as thin and spines with a defined neck and a prominent head are classified as mushroom. Filopodia were few in number across all samples and were excluded from analysis. For the calculation of surface NMDAR/AMPAR ratio, ROIs were defined on dendrites from expressing neurons (SA-Ch/palmitoyl-Turquoise2 or palmitoyl-Turquoise2), and mean sGluR1/Alexa488 and sGluN1/Alexa647 intensities were calculated after background subtraction. The calculated value is the ratio of the two means.

EI was calculated as the ratio between the Cherry average intensity on the spine region (identified using the EGFP channel) and the average intensity calculated on the dendritic shaft between 1 and 2 µm away from the spine junction, after background subtraction. For the EI calculation, only expressing spines were included in the analysis. Homer1c-EGFP content was quantified by integrating EGFP intensity in correspondence to the PSD and normalized by the mean intensity on the dendrite. SEP-GluA1 enrichment was calculated in an analogous manner to Cherry EI; for the comparison of the two EIs, the same regions were considered in the two channels.

For two-photon stimulation experiments, spines were identified in the EGFP filler channel, Cherry fluorescence was integrated in the corresponding channel after background subtraction. Intensity was calculated for images acquired immediately before photouncaging and after 60′ for stimulated and neighboring spines. The relative change in Cherry intensity ($I_c$) at time point $i$ was calculated as the difference, normalized for the initial intensity as follows: $[I_c(i) - I_c(-5')]/I_c(-5')$. Volume change was calculated in an analogous way as $[I_g(i) - I_g(-5')]/I_g(-5')$, where $I_g$ is the spine integrated density in the EGFP channel, normalized by the mean value in the dendrite underneath.

For the intensity profiles shown in Supplementary Fig. 10, 1,024 × 1,024, 0.5 µm stacks of immunostained slices were acquired by centering the field on the CA1 region above the DG upper blade. For each channel, slices were summed to generate the projection image. After background subtraction, linear profiles of 325 µm (80 µm thick) were measured starting from the stratum oriens toward the stratum lacunosum-moleculare in correspondence to all detected EGFP-positive neurons. Profiles were aligned in the DAPI channel by setting the start of the stratum pyramidale, identified as the stratum with packed soma, at 100 µm. For EGFP and Cherry channels, baseline was subtracted and resulting profiles were averaged. Baseline was evaluated in the non-electroporated hemisphere in an analogous manner. To reduce noise, resulting data were smoothened with SigmaPlot v12 (SYSTAT) with the median method (0.01 sampling) in Supplementary Fig. 10b. To compare SA-Ch profiles and Thy1:ChR2-YFP profiles, first values were averaged in a 5 µm window. Resulting profile data were averaged

and normalized on the highest value, and two-way analysis of variance (ANOVA) was performed (factor A: SA-Ch/Thy1-YFP, factor B: distance).

For in vivo experiments, spine distance was calculated as the euclidean distance as $[(x_1-x_2)^2 + (y_1-y_2)^2 + (z_1-z_2)^2]^{1/2}$ where (x,y,z) are the spine coordinates in micrometers. Distances were calculated for each SA-Ch-positive spine to all other SA-Ch-positive spines and all SA-Ch-negative spines with a custom-made program in R (version 3.3.1, available at http://www.R-project.org). Distance to first potentiated neighbor and to first non-potentiated neighbor are defined as the minima of the two sets, respectively. The distances to first potentiated and first non-potentiated neighbor were also calculated for randomly shuffled data by using the "sample" module in R to randomly assign the identity of spines to the (x,y,z) positions; for every dendrite, five shuffled datasets were considered.

Probability data were calculated as follows with a custom made program in R: for each SA-Ch-positive spine we considered the first 20 neighbor spines (both directions along the dendrite were considered). For each dendrite, this gave a set of N sequences of 20 spines that could be aligned from position 1 to position 20 generating a N×20 matrix. For each $i$-th column, we counted the number of positive spines and divided it by the number of rows N. The resulting value is the probability of finding a potentiated spine in position $i$. To calculate the increase in probability, for each dendrite probability data were divided by the expected probability of finding a potentiated spine in the corresponding position if they were randomly arranged. Thus, calculated values were divided by $(p-1)/(T-1)$, where $p$ and $T$ are the number of potentiated spines and the total number of spines in each dendrite, respectively.

Clusters of SA-Ch-positive spines were calculated with the "Hierarchical cluster analysis" function in R (contributed to STATLIB by F. Murtagh) with the "single linkage" method with a 2 µm threshold. Thus, two spines belong to a cluster if their distance (calculated as above) is lower than 2 µm. Together, spine clusters and single spines (spines that do not have another SA-Ch-positive spine within 2 µm and could therefore be regarded as cluster with dimension = 1) constitute spine units. Accordingly, spine unit dimension was calculated as the number of members for each unit.

To calculate the SI, the average fraction of potentiated spines ($f$) was calculated for each slice as the sum of SA-Ch-positive spines in dendrites belonging to the slice divided by the total number of spines. Then, for each dendrite, the expected number of potentiated spine $p^*$ was calculated as $f \times T$, where $T$ is the number of spines in the dendrite. SI was calculated as the absolute value of $(p-p^*)/p^*$, where $p$ is the number of potentiated spines in the dendrite.

**Statistics**. Image analysis was performed using ImageJ. Statistical analysis was performed with OriginPro v9.0 or GraphPad Prism 6. Differences between two groups were evaluated with two-tailed Student's $t$-test. Residues (Supplementary Fig. 3) distributions were compared with Kolmogorov-Smirnov test. Multiple comparisons were made by one-way ANOVA followed by post-hoc Bonferroni test, unless otherwise stated. Significance was set at $\alpha = 0.05$. One hundred and thirty neurons were analyzed in Supplementary Fig. 2a–c; 106 dendrites from 35 neurons were analyzed in Supplementary Fig. 2d. Seven hundred and fifty-six frames (21,564 spines) were used for ChETA constructs expression calculation. A total of 1,493 spines was analyzed for EI calculation. For PSD95/ChETA-Cherry co-localization, a total of 2,251 spines from 44 neurons were analyzed. For Homer1c-EGFP/SA-Ch correlation, 369 spines from 37 neurons were analyzed; for SEP-GluA1 experiments, 737 spines from 108 neurons were analyzed.

For two-photon uncaging experiments, a total of 48 samples were analyzed, and a total of 118 spines were considered. For the time-course experiments in two-photon uncaging, we considered the following number of spines: uncaging, 18 stimulated and 24 nearby spines; uncaging with anysomicin, 15 stimulated and 21 nearby spines; without MNI-caged glutamate, 8 stimulated and 8 nearby spines.

For GCaMP6s imaging, 55 spines were stimulated, out of which 17 in presence of VGCC inhibitors and 17 in the presence of TTX. For in vivo analysis, the following number of dendrites (spines/slices/animals) were considered: CA1 home cage 93 (6703/8/4), DG home cage 52 (4157/9/4), CA1 novel context 111 (10223/8/3), and DG novel context 58 (4865/8/3). Thirteen dendrites were excluded from the calculation of the increase in probability according to pre-established criteria, because (i) the fraction of positive spine was below the defined threshold of 0.05 or (ii) it was not possible to define a whole set of 20 neighbors. Data were analyzed from two researchers; spine notation was performed by a blind researcher to condition (home cage/context) but not to hippocampal region (CA1/DG). No statistical method has been used to pre-determine sample size. Animals were distributed randomly between groups. Animals where the hippocampus was not electroporated (due to electrodes misalignment during electroporation) were excluded from the analysis, as established before the analysis ($n = 1$).

Comparisons between groups when distributions were not normal, or requirements for parametric tests were not met, were performed with Kruskal–Wallis test, followed by Dunn's test for pairwise comparison. Non-parametric comparisons between two samples were performed with Mann–Whitey test. Parametric test used were Student's $t$-test (two-tailed) (paired tests were used to compare sets of coupled values in Fig. 4b), or Bonferroni correction following one-way ANOVA for multiple comparisons.

All information is summarized in Supplementary Table 1.

**Data availability**. Data supporting the findings of this study are available within the paper and its supplementary information file. Reagents will be made freely available upon request and the exchange of materials will be regulated by an MTA. All R codes are freely available upon request.

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

## Acknowledgements

We thank Dr S.Kindler (Uni Hamburg) for useful discussion and for providing MAP2 and alphaCaMKII cDNA, to amplify DTEs, Dr A. Riccio (UCL) for IMPA1 ATE cDNA, Dr A. Marcello (ICGEB Trieste) for plasmids pSL-MS2 12X and Cherry-MS2 coat protein-NLS, and Dr D. Choquet (CNRS Bordeaux) for making plasmid Homer1c-EGFP available. We thank G. Testa (Scuola Normale Superiore, Pisa) for help with animal perfusion and slice preparation for IF. We thank C. Rizzi and N.M. Carucci (Scuola Normale Superiore, Pisa) for help with cortex and hippocampus dissections. The work was financially supported by institutional funding from Scuola Normale Superiore and by the E.U. flagship Human Brain Project (grant number 604602).

## Author contributions

A.C. conceived the project. F.G. and A.C. designed research. F.G., L.M., A.J., B.P., N.B., F.P.B. and C.A. performed experiments and analyzed data. M.C. supervised and provided expertise on the optogenetic experiments in culture. S.L. supervised two-photon uncaging experiments. T.F. supervised and provided expertise on the measurement of synaptic glutamatergic transmission in *ex vivo* hippocampal slices. L.C. supervised and provided expertise on the *in utero* electroporation. A.C. provided expertise and discussion on the in utero data. F.G., L.M., S.L. and A.C. wrote the manuscript.

## Additional information

**Competing interests:** The authors declare no competing financial interests.

