## [Peer Review File · Nature Communications]

Reviewers' comments:

Reviewer #1 (Remarks to the Author):

In this manuscript, the authors developed a novel tool which enables the expression of an opsin to active dendritic spines, using a combination of trafficking signals. This tool might enable a direct mean to activate or inhibit active spines. If the strategy works in vivo, this tool could help neuroscientists probe a subset of spines which are learning related and possibly test the necessity of these spines for specific learning events, or either examine specificity of memory ensemble in the level of single spines, during in vivo experiments.

To achieve this goal, the authors took a strategy similar to a recent paper from Japan (Hayashi-Takagi et al., Nature 2015): they insert a genetic tag utilizing Arc UTR elements and a post-synaptic targeting peptide to a variant of channel rhodopsin 2 (ChR2). Thus, the strategy is designed to provide local protein synthesis-dependent synaptic targeting of opsin. Following testing and validating of this Arc-SYP tag, they examine utilization of opsins to activate targeted spines in dissociated neuronal cultures.

While the strategy is interesting and potentially useful for in vivo work, the authors did not provide any evidence that their probe could be useful in in vivo situation for study of learning and memory. Most of experiments are done in dissociate culture with non-physiological stimulation.

Thus, overall the novelty and significance of the study are limited.

Major points

1. One of the important characteristics of ArcSYP-Ch is that the expression appears to be correlated with the spine volume or PSD size, which are likely correlated with function of the spine. However, the correlation was not quantitatively measured. If the correlation is simply linear, it is not actively accumulated to the high functional spines and thus not useful. If the correlation is super-linear, it may be useful but needs to be carefully quantitated. In addition, to evaluate the correlation of ArcSYP-Ch expression with function, it is important to measure superecliptic pHluorin-GluA1 as well as the volume (or electrophysiological response by glutamate uncaging from ChETA-positive and negative spines).

2. It is not evaluated what kind of plasticity is correlated with the expression of ArcSYP-Ch in spines in realistic situations. In Takagi-Hayashi's paper (2015), the expression of AS-Rac was clearly correlated with spine growth and spino-genesis during motor learning in vivo. However, the authors tested mostly with global stimulation (like BDNF application, chemical LTP etc) in dissociated culture. To have a significant impact to the field, it would be necessary to show that the probe/actuator works in vivo or at least in slices.

3. One experiments was performed to test spine-specificity using glutamate uncaging in combination with forskolin. However, in this experiment it is not clear what kind of plasticity is induced, since they measured only one time point (60 min) but not full time courses (like most of previous experiments did). If this is protein-synthesis dependent plasticity, it should last several hours and with slower rising phase. Also, does spine volume change occur in all

stimulated spines (perhaps not)? If not, is it correlated with the accumulation of SYP-Ch? Does this paradigm produce functional plasticity? It is necessary to measure the time course of spine structural plasticity and electrophysiological potentiation (or maybe SEP-GluA1) to see the correlation between the expression of ArcSYP-Ch and structural/functional plasticity of spines.

4. Glutamate uncaging experiments requires control with protein translation inhibitor right before application of uncaging to show that these are newly synthesized proteins induced by glutamate uncaging. In addition, does Arc promoter do anything on this paradigm? This point needs to be clarified with transcription inhibitors.

5. The main emphasis and novelty in this paper is combining and harnessing this probe for labeling active spines, with opsin light activation. For this end, the authors examine Calcium transients, using GCaMP6s, to enable direct measurements in labeled spines. However, the experiments appear to be preliminary. They stimulate opsin with bright field illumination while imaging with the same illumination: thus, the indicated fluorescence is in arbitrary unit instead of $\Delta F / F$, a usual measure of Ca^{2+} that is relatively independent of volume and concentration. Because of this, it is not possible to quantify the amplitude and kinetics of synaptic activation occurs in these spines. The authors need to use different fluorophore/opsin combination to separate the imaging and photoactivation to quantify Ca^{2+} level in response to more defined laser pulses (in in-vivo situation, perhaps trains of short pulses will be used). In addition, it will be useful to measure electrophysiological response from single spine and global activation to figure out what is the current produced in each spine and how many spines are activated.

6. For the whole neuron illumination, there seems to be a significant current from the thick dendrite or the soma. For in vivo experiments, this may be a huge draw-back if one wants to activate only a subset of spines that is related to memory but not the whole neuron. Thus, it will be important to measure the degree of somatic activation compared to spines. I think this can be only done with electrophysiological recording.

7. Related to 6, C-fos in Fig. 7 staining is perhaps not due to the activated spines, but more likely to be caused by direct activation of the soma. The authors need to show the benefit of spine-specific activation in this context.

8. P-CaMKII staining --- one of the most beautiful P-CaMKII staining I have seen. However, it requires control of antibody in side-by-side control with either CaMKII KO or ideally CaMKII T286A mice.

Minor points:

Line 36: "aninput" should be "an input"

Fig. S3A +BDNF Images: they contain irregular gray rectangles.

Fig. S3A +Saline image: the order of images appears to be wrong.

Fig. S7A: it appears that there are two gray crosses, but not explained in the legends what they are.

Reviewer #2 (Remarks to the Author):

Recommendation: This paper has some nice ideas, but in its current form is too messy to publish. It was actually quite hard to review. The reasoning for making ArcSYP-ChR2-Cherry in the first place is not even clear? How would such a tool be used? The use of the word "engram" implies *in vivo* use, but I can't see how such a tool would be useful *in vivo*. It might be more spine-enriched than cytoplasmic ChR2, but the massive soma expression is going to lead to neural spiking and back-propagating action potentials, which will negate the supposed spine-specificity of the manipulation. Is this a slice tool? Cultured neurons? The *in vivo* portion of the Discussion is unconvincing.

Major things:

- A lot of statements are unnecessarily made too strong. For instance, "No causal role has yet been demonstrated for the modifications of synapses in the formation of memories." Really, a great number of people would be surprised to hear this claim. Of course synaptic modifications have been shown to be critically involved in memory formation, strengthening, recall, etc. But there are many details yet to be worked out...
- "Indeed, the current spatial resolution of opsin expression is limited to whole cells but does not allow selective subcellular localization control." This statement is patently false. Opsins can and have been targeted to the axon initial segment, to the soma, to the mitochondrion, to dendrites, to synaptic vesicles, etc. Some of this work has been done a decade ago. I really have no idea what the authors mean by this. The statement could not be more incorrect. The authors then go on to cite some of these papers but say that they lack sufficient specificity.. ???
- the authors should cite some of the papers on "single-synapse optogenetics." Some of those have been very useful.
- I think the data provided by the authors do not support the claim "Arc RNA sequences determine the uneven tagging of synapses, while the SYP tag docks the protein inside the synapse." The data in the paper are kind of all over the place on those points.
- the images that contain a cell body make it clear that the vast majority of the ChETA, with all the targeting motifs, remains in the soma. The authors make such a big deal about needing it not to be expressed there that it's surprising that they don't comment on this.
- one thing that's very confusing is that the authors don't do a good job distinguishing between the fact that ChETA-mCherry (even with the targeting motifs) seems to preferentially accumulate in large spines over small ones at baseline, and then the fact that GCaMP6 signal is stronger from the large spines. Maybe they underwent LTP, or maybe they just expressed more ChETA and/or GCaMP6. The correspondence between spine size and LTP/synaptic strength is incompletely understood.
- far too many details are left out for even an expert to reconstruct how the experiments and data analysis were done. What were the lookup tables for GCaMP signal binning? Those images look too clean as it is right now- I think the lookup tables have been adjusted to show the desired outcome. What laser powers were used? What excitation and emission filters?
- the writing is rough throughout. I had to reread some sections 3-4 times to figure out

what was going on.

Medium things:

- In Fig. 1Aii-ii', the authors say that mCherry was quite uniform along dendrites, when instead it is obviously blebby, as typically happens with dendritic expression of mCherry.
- the authors say that "SYP-Ch was evidently enriched at spines compared to unmodified ChETA-Cherry", but that doesn't seem clear from the images.
- also, they say that large synapses preferentially express Arc-Ch over smaller ones, but there really is no quantification. The images are pretty bad, too, which doesn't help interpretation.
- the authors say that the Arc DTE is "the best candidate" for localization, but then in Figure S1 they show that CaMKII and MAP2 are 3 times better than Arc. Very confusing.
- Figure S4 fairly clearly shows that after KCl treatment, both ArcSYP-ChR2 mRNA and protein are preferentially localized to the soma. The whole point of the paper is that this should localize to spine heads.
- the flux through ChR2 is probably a negligible source of spine $[Ca^{2+}]$. Depolarization of the spine head will open voltage-gated Ca^{2+} channels, that are going to be the bulk of the source of spine $[Ca^{2+}]$.
- in Figure 4, it's not clear how the ChR2 is being activated. It is being activated by the GCaMP6 imaging light? Or is a separate light source illuminating it?
- In Fig. 4A, only 2 of the panels show the soma, so it's impossible to tell how well the targeting worked. The authors say, "The expression pattern of Arc-palmitoylCherry was very similar to ArcSYP-Ch (and Arc-Ch) (Figure 4A)." I thought their whole claim was that ArcSYP targeted things to spines. And instead of "Arc-Ch", do they mean "SYP-Ch"?
- the spine GCaMP6 signals are harder to interpret than the authors let on (or perhaps realize). From lots of experience using it, smaller spines tend to show higher DF/F than larger ones. The reason almost certainly is that as Ca^{2+} comes into a much smaller volume, the local concentration increases picked up by GCaMP6 are much higher. And since the Hill coefficient of GCaMP6 is >3 , this translates into a much greater photon flux from fewer highly-saturated GCaMP6 molecules than from more less-saturated GCaMP6 molecules. It seems that they're largely comparing the ratio of GCaMP signal at spines to shafts, but they might need to get into a discussion of how absolute GCaMP signal is not reflective of spine strength.
- what is the source of the GCaMP6 signal in the cells in TTX and not expressing ChR2?
- when the authors say, "even single inputs could drive significant localized inputs that were not overcome by the activation of the whole rest of the neuron (Figure 4D)", then doesn't that throw into question the whole point of trying to focus on spine-targeted ChR2?

Minor things:

- the authors don't define G418 as geneticin.
- the labels seem to be swapped for Fig. 2D.
- line 525: "vi" should be "iv"

Reviewer #3 (Remarks to the Author):

The manuscript by Gobbo et al., titled: "Activity-dependent expression of Channelrhodopsin in neuronal synapses" describes a novel targeting strategy for optogenetic tools, designed to express a light-gated ion channel specifically in previously-activated synapses. To target the opsin into the synapses, the authors utilized a clever duplex strategy combining RNA- and protein-targeting sequences that achieve very high specificity to post-synaptic sites and demonstrate robust activity-dependence. Addition of the immediate-early gene promoter Arc modulates translation of the so targeted mRNA in an activity-dependent manner. The imaging and immunostaining data convincingly demonstrate synapses-specific targeting as well activity-dependent expression. The authors then use calcium imaging to evaluate the efficiency of stimulation of ChETA-expressing synapses with light, quantifying GCaMP6 fluorescence and phosphorylated CaMKII as proxies for induced synaptic currents.

Overall, this article has tremendous innovative potential and I am certain this work will be interesting for many neuroscientists. I believe that with a few additional experimental controls as detailed below, this manuscript would be highly suitable for publication in Nature Communications. I would be more enthusiastic if the authors find a more a convincing way to quantify the efficiency of reactivation of synapses with light. Inclusion of electrophysiology experiments will allow a more precise quantification of light-evoked currents and comparison with real synaptic currents in the same neurons. Some additional experimental data is missing, and its inclusion would facilitate reproduction of these experiments by other labs.

Specific comments:

1. My major concern is that it's unclear whether GCaMP6s fluorescence changes are due to calcium influx through ChETA. This variant does not have an especially high calcium conductance and is engineered for short opening time and therefore conducts in absolute terms the least amount of ions compared to other ChRs. The paper cited in reference 41 actually doesn't contain any data relating to ChR2 as calcium-permeable. Reference 42 demonstrates a detectable ChR calcium influx at 80mM extracellular calcium. This is very common in the community (for a good reference, see Schneider et al., Biophysical Journal 2013), but I am not aware of any published data set in which such strong calcium-influx through ChR was detected at physiological conditions. I argue that the effect the authors see is due to ChR induced voltage-deflection which partially open high conductance voltage-gated calcium-channels. Therefore, an experiment with nifedipine, barium or cadmium would answer this question.

2. The experiment in Fig 4 A are difficult to interpret since ChETA and GCaMP are excited in parallel – so a stable baseline is missing. A cleaner approach would be to use red-shifted genetically encoded calcium-indicator such as jRCaMP or a red-shifted chemical dye. Again I am very doubtful that the transient influx upon illumination is really due to higher conductance of the O1 state. Since the authors seem to be able to use a two-photon microscope, combining two-photon calcium imaging with full-field light stimulation might

also allow a stable calcium baseline.

3. The targeting sequence utilizes an NMDA consensus sequence as well as a PSD95 binding site. Utilizing these sequences might alter intrinsic electrophysiological properties of the synapse by competing with endogenous binding partners of PSD95. A rough estimate of the number of targeted ChRs ($35\text{pS}/\mu\text{m}^2$ Grossmann, J. Computational Neuroscience 2013 or Zimmermann et al, 2006, Biochemical and Biophysical Research Communication), and an average PSD95 area of $.02 - 0.3 \mu\text{m}^2$, the expected voltage deflection will be rather small. I still believe that it is possible to mimic postsynaptic potentials, I would just urge the authors to provide electrophysiological data which will more clearly demonstrate what kind of responses are induced as well as whether endogenous synaptic properties are changed. These experiments will provide the necessary strength which will make this work more convincing for a lot of readers.

4. In figure 1C, it seems that Arc enrichment is not significantly different from with BDNF, but for the spine count, measured as "fraction of tagged spines", the BDNF treatment increases the number of spines drastically. Does this mean that Arc only increases expression in previously non-tagged spines and is not actually increasing the expression level of already expressing spines?

5. From the figures, it appears that ChRs are also expressed to substantial amounts at the neuronal soma. This expression should be measured and quantified in comparison with non-targeted channelrhodopsins. Sufficient functional expression of ChR in the soma could explain the c-fos data. Furthermore, under more natural conditions, in which spikes can be readily generated at the soma and back-propagate to the dendrites, the effect of the activation of specifically targeted ChRs in the spines might be overwhelmed by the somatic spiking driven by this non-specific expression. How do the authors intend to overcome this limitation when this tool is applied in vivo? Perhaps an addition of a somatic destabilization sequence would bias the turnover rate of somatic channels?

6. Fig 5 C is a very convincing experiment – the mCherry negative spines (empty boxes) are analyzed from transfected neurons. Does this imply that only spines which had neighboring mCherry positive spines were taken into account in the analysis?

Minor points:

Line 244: Please specify which sequences you tested and add them in the supplements

Line 44: What is this minimal sequence – please present it in the supplements. The reference to citation number 20 is confusing since it implies that the minimal sequence of Arc is published there. I could not find this information in this reference.

Line 19: Channelrhodopsin is an opsin, but not all opsins are ion channels.

Line 20: The Malinow paper specifically modulates synaptic inputs to the amygdala – it is given as ref 7 in your manuscript. You might want to rephrase to "modulating specific

synapses". The same applies to the statement in line 220 – there are in fact a few papers describing synapse-specific manipulations.

Line 36: space missing between "aninput"

Line 42: Please provide the reference for the "fast maturing mCherry".

Figure legend S1: Please change "I is ..." to something like "Intensity (I)" – it is a bit confusing.

Figure legend S1: Please provide further information regarding how many cells and how many preparations are included in the data set. Similar to the information you provide in the spine data set in the Statistics part of the manuscript.

Line 72: "Conversely, SYP-Ch distribution was quite similar to the EGFP one under the same conditions.". This sentence is confusing, since it is not really the same conditions. isn't it better to call it (-) BDNF?

Line 85: Is the comparison depicted in Figure 2A significant?

Line 92: "smaller spines were not as effectively labelled as the dendrite (Figure 1B)" – this statement is meaningless without some quantification of the imaging data.

Line 126 – Could it be that the expression of ArcSyp-Ch is actually increasing size of synapses? Does Arc-Syp-Ch and Arc-Ch constructs target different sized synapses after excitation with one of the chemical LTP protocols (Figure 1 D). This might indicate that Arc-Syp-Ch actually causes the increase in synapse size.

Line 134 – Why did you include Forskolin in the extracellular medium? Is forskolin required for activity-dependent expression?

Fig 3 – The switch in terminology from Arc-Syp-Ch to AS-Ch is confusing – it occurs a few times in the different figures. Also, "pre" and "post" in the context of synapses is misleading – maybe better use "before" and "after" as used in the text

Also regarding nomenclature: the full construct is "N-ChETA-mCherry-SYP-C", and in this case I find it a bit confusing to abbreviate it as SYP-Ch – please consider revising the nomenclature.

Based on the schematic drawing in Figure 5 A and S6 – the experiments are performed less than a day after transfection. How does the expression pattern look after 2 – 3 days? Are there more synapses labeled? Do the authors see a similar increase of tagged synapses as shown in figure 1D? This would be highly important for in vivo studies.

Can the authors provide the exact acquisition parameters for the calcium imaging –light intensity, frame resolution – it is a difficult experiment! Was frame size always the same?

Figure 3 – to evaluate the size of synapses the authors use the mCherry fluorescence – I don't think it is that straight forward to transfer the theory of a diffusible filler of Yasuda (ref 37) to a membrane protein. Especially when the construct is supposedly anchored to PSD95. Do you actually see an increase in size of the fluorescence or is it only intensity that increases?

What are the absolute expression levels for Arc-Ch and Arc-Syp-Ch? If Arc-Ch is significantly lower or higher expressed, a non-linear response curve for the photodiode could lead to the difference of relative fluorescence ratios.

A membrane-bound Arc-Syp-Palm-mCherry construct as could have been a better control to see whether expression of ChR is actually changing size of synapses.

Figure S3 – the scale bars in the images are 10µm while the profile plot is 200µm – but the image is not 200µm large. I found it confusing since it indicates that this is the part of a larger image which was used to analyze and create the graph.

Figure 1D – the title "spine count" is confusing – maybe label axis "fraction of tagged spines"?

We thank the reviewers for their in depth analysis and useful comments. We hold them in great consideration, and we modified the paper accordingly. We addressed all the concerns that were raised and we incorporated new experiments that respond to the Reviewers' suggestions. We have answered as systematically and completely as possible to the Reviewers' requests, and we believe that all main issues have been addressed. We thank the Reviewers and the Editor for their constructive help, and we think that thanks to their comments the paper has significantly improved. We provide point-by-point response to the Reviewers' questions below; for clarity, we attach the original Reviewers' comments in blue and we provide comments below in black. In the responses, at the end of each paragraph, the Reviewers can find indicated within square brackets ([###-###]) the line number(s) where the issue has been specifically addressed in the revised manuscript; line number preceded by S is referred to the Supplementary Information file.

Please note that, as suggested by Reviewer #3, we changed the nomenclature of the constructs as follows: Arc-Ch is now A-Ch, ArcSYP-Ch is now SA-Ch, and SYP-Ch is now S-Ch. We will be using this new nomenclature throughout this letter for consistency with the modified nomenclature in the text. Instead, we left the names unchanged in the text of the Reviewers' questions, as well as when directly referring to specific statements of the Reviewers.

Reviewer #1 (Remarks to the Author):

In this manuscript, the authors developed a novel tool which enables the expression of an opsin to active dendritic spines, using a combination of trafficking signals. This tool might enable a direct mean to activate or inhibit active spines. If the strategy works in vivo, this tool could help neuroscientists probe a subset of spines which are learning related and possibly test the necessity of these spines for specific learning events, or either examine specificity of memory ensemble in the level of single spines, during in vivo experiments.

To achieve this goal, the authors took a strategy similar to a recent paper from Japan (Hayashi-Takagi et al., Nature 2015): they insert a genetic tag utilizing Arc UTR elements and a post-synaptic targeting peptide to a variant of channel rhodopsin 2 (ChR2). Thus, the strategy is designed to provide local protein synthesis-dependent synaptic targeting of opsin. Following testing and validating of this Arc-SYP tag, they examine utilization of opsins to activate targeted

spines in dissociated neuronal cultures.

While the strategy is interesting and potentially useful for *in vivo* work, the authors did not provide any evidence that their probe could be useful in *in vivo* situation for study of learning and memory. Most of experiments are done in dissociate culture with non-physiological stimulation.

Thus, overall the novelty and significance of the study are limited.

We thank Reviewer #1 for constructive criticisms and comments; we did our best in trying to answer the mentioned issues and in particular we are confident that the novelty, scope and significance of the paper have been improved.

Major points

1. One of the important characteristics of ArcSYP-Ch is that the expression appears to be correlated with the spine volume or PSD size, which are likely correlated with function of the spine. However, the correlation was not quantitatively measured. If the correlation is simply linear, it is not actively accumulated to the high functional spines and thus not useful. If the correlation is super-linear, it may be useful but needs to be carefully quantitated. In addition, to evaluate the correlation of ArcSYP-Ch expression with function, it is important to measure superelectaptic pHluorin-GluA1 as well as the volume (or electrophysiological response by glutamate uncaging from ChETA-positive and negative spines).

We agree with Reviewer #1, and we correlated SA-Ch expression to the PSD size. Diffraction-limited resolution does not allow absolute spine volume quantification, so we evaluated the PSD size by Homer1c-EGFP intensity. Homer1c is part of the PSD whose enrichment in the spine was found to be an excellent indicator of spine volume (Meyer et al. Neuron 82, 430–443, 2014). As shown in the new Figure 2e, SA-Ch negative spines displayed a smaller PSD size (evaluated as Homer1c-EGFP content), and the SA-Ch/Homer1c-EGFP plot for the spines is clearly super-linear. [128-132]

As for the point raised on GluA1 receptors, we also found a significant correlation between SA-Ch and SEP-GluA1 enrichment when they were co-expressed in neurons (Figure 2h,g). SA-Ch was limited to SEP-GluA1 expressing spines, in contrast to S-Ch, whose expression is not regulated. [132-139]

2. It is not evaluated what kind of plasticity is correlated with the expression of ArcSYP-Ch in spines in realistic situations. In Takagi-Hayashi's paper (2015), the expression of AS-Rac was clearly correlated with spine growth and spino-genesis during motor learning *in vivo*. However, the authors tested mostly with global stimulation (like BDNF application, chemical LTP etc) in dissociated culture. To have a significant impact to the field, it would be necessary to show that the probe/actuator works *in vivo* or at least in slices.

In response to this point, we have performed a significant amount of new experiments, demonstrating that the SA-Ch protein is expressed *in vivo* in a synapse-activity dependent manner. The plasmid encoding for the SA-Ch protein was electroporated *in utero* and the expression of the SA-Ch protein was studied in the hippocampus, after transcriptional induction with doxycycline and exposure of the mice to a novel context. Indeed, in mice and rats Arc is known to be expressed in the hippocampus following the exposure to a novel context or LTP (Minatohara et al. Front Mol Neurosc, 2014; Steward et al Front Mol Neurosc, 2015). Analogously, c-fos expression is activated following a novel context exploration. Thus, novel context exposure represents a powerful natural stimulus for hippocampal neurons. We evaluated the response of SA-Ch *in vivo* to the exposure of a novel context, finding a significant increase in the number of labeled spines in CA1 and DG dendrites (new Figure 5). We detected clusters of potentiated spines, that increase in dimension (number of spines per cluster) between the animals exposed to a novel context and those that were kept in their cage. [214-241]

These new data represent a good demonstration that the SA-Ch protein works *in vivo*, as the Reviewer asked.

3. One experiment was performed to test spine-specificity using glutamate uncaging in combination with forskolin. However, in this experiment it is not clear what kind of plasticity is induced, since they measured only one time point (60 min) but not full time courses (like most of previous experiments did). If this is protein-synthesis dependent plasticity, it should last several hours and with slower rising phase. Also, does spine volume change occur in all stimulated spines (perhaps not)? If not, is it correlated with the accumulation of SYP-Ch? Does this paradigm produce functional plasticity? It is necessary to measure the time course of spine structural plasticity and electrophysiological potentiation (or maybe SEP-GluA1) to see the correlation between the expression of ArcSYP-Ch and structural/functional plasticity of spines.

For the experiment that the Reviewer refers to, we employed a published protocol (Hill and Zito, *J. Neurosci.*, 33, 678–686, 2013). To confirm our data and answer the reviewer’s criticism, we repeated the experiment by taking multiple time points up to 90 minutes following spine stimulation. Volume (evaluated by EGFP intensity) rapidly increased and reached a plateau that lasts for the whole considered time, whereas the increase in SA-Ch increased over time and reached a plateau after about 60 minutes. This likely reflects the time course needed for (i) translation to occur and (ii) newly-synthesized Cherry to mature. The temporal profile was similar for all analyzed spines, and only one spine out of 18 returned to a level close to the initial one. The induced plasticity was translation-dependent, as translational blockage with anisomycin prevented volume change to be maintained. Data are reported in the new Figure 3d_{1,3}. [151-152; 157-161]

4. Glutamate uncaging experiments requires control with protein translation inhibitor right before application of uncaging to show that these are newly synthesized proteins induced by glutamate uncaging. In addition, does Arc promoter do anything on this paradigm? This point needs to be clarified with transcription inhibitors.

We followed Reviewer #1’s suggestion and repeated our experiments in the presence of anisomycin. Translation inhibition blocked SA-Ch accumulation at stimulated spines. Data are reported in Figure 3d_{1,3}. [160-161]

As for the question on Arc promoter, we don’t think Arc promoter to play a significant role, since the expression of SA-Ch transcript is not driven by Arc promoter, but by the constitutive CMV promoter. In addition, we think that the time course of SA-Ch accumulation would preferentially support a role for de-repression and translation from pre-existing RNA, as Arc transcript is initially exported after 15 minutes from the stimulation onset, and transcription inhibitors typically exert their effect on a longer time scale (100-150 minutes, Kelleher III et al, *Neuron*, 44, 59-73, 2004). [341]

5. The main emphasis and novelty in this paper is combining and harnessing this probe for labeling active spines, with opsin light activation. For this end, the authors examine Calcium transients, using GCaMP6s, to enable direct measurements in labeled spines. However, the experiments appear to be preliminary. They stimulate opsin with bright field illumination while imaging with the same illumination: thus, the indicated fluorescence is in arbitrary unit instead of $\Delta F / F$, a usual measure of Ca²⁺ that is relatively independent of volume and concentration. Because of this, it is not possible to quantify the amplitude and kinetics of synaptic activation occurs in these spines. The authors need to use different fluorophore/opsin combination to separate the imaging and photoactivation to quantify Ca²⁺ level in response to more defined laser pulses (in in-vivo situation, perhaps trains of short pulses will be used). In addition, it will be useful to measure electrophysiological response from single spine and global activation to figure out what is the current produced in each spine and how many spines are activated.

We understand the point underlined by the Reviewer #1, and we modified the experimental setup accordingly. To be able to register a stable baseline, we followed Reviewer #3’s advice to excite GCaMP with two-photon illumination. SA-Ch was excited with a brief illumination (10ms) at 488 nm by means of spiral scanning in a region centered on the spine. GCaMP imaging was performed in a non-overlapping region at 990 nm. In this way we were able to express Ca²⁺ fluorescence in terms of $\Delta F / F$; we therefore recorded Ca²⁺ influx following 488nm light stimulation. We were able to separate GCaMP imaging from Channelrhodopsin excitation because our imaging region did not contain the PSD region, where the vast majority of Channelrhodopsin is expressed. Further details are provided in the revised text in the Materials and Methods section, and the resulting experiment is presented in Figure 4. [167-175; 397-410]

6. For the whole neuron illumination, there seems to be a significant current from the thick dendrite or the soma. For in vivo experiments, this may be a huge draw-back if one wants to activate only a subset of spines that is related to memory but not the whole neuron. Thus, it will be important to measure the degree of somatic activation compared to spines. I think this can be only done with electrophysiological recording.

We agree with Referee #1 that the ratio of synaptic versus somatic expression is a crucial parameter to be characterized. Our new in vivo expression experiments, following in utero electroporation of the SA-Ch plasmid, show that, in the hippocampus, SA-Ch is largely excluded from somas, while expressed at synapses along basal and apical dendrites. This is qualitatively shown in the images in Fig 5 and quantified in Figure S7, by plotting Cherry intensity profile of CA1 neurons; we took advantage of the anatomical organization of CA1 pyramidal neurons, with somas densely packed in the stratum pyramidale and dendrites mostly confined in the upper and lower layers. Thus, in vivo, we find that Cherry fluorescence is most prominent in the dendritic layers. This finding settles in a convincing way the issue of the somatic versus synaptic expression in vivo. As for the in vitro situation, we regard the residual somatic expression of SA-Ch observed in dissociated cultures as a phenomenon due to overexpression: (i) the promoter used in the in vivo experiment is a weaker one than that used in the in vitro experiment and (ii) the dendritic arborization of neurons grown in culture is significantly smaller than the one of in vivo neurons. Therefore an excess of transcribed RNA may saturate dendritic transport and the associated translational repression system, thus leading to leaking somatic translation. Under in vitro overexpression conditions, it is therefore more likely for some RNA molecules to escape the molecular

machinery of the neuron that prevents them from being translated. In addition, somatic translation in culture was most evident in stimulated neurons, with respect to unstimulated conditions; however, as Reviewer #1 noted, stimulations performed in cultured neurons are a global treatment that can signal the transcript to be derepressed also in the soma. Thus, under in vivo conditions the degree of somatic activation compared to that in spines is very favourable, also because the stimulation that CA1 neurons received was a natural one. [114-118; 209-220; S71-S77]

7. Related to 6, C-fos in Fig. 7 staining is perhaps not due to the activated spines, but more likely to be caused by direct activation of the soma. The authors need to show the benefit of spine-specific activation in this context.

We cannot exclude that the direct activation of the soma might contribute to c-fos expression (in the present Figure 4), but we consider this unlikely, as cultures that received the light stimulation protocol were not exposed to previous stimulation to induce SA-Ch expression. Under these conditions, residual somatic expression in mature neurons (day in vitro 14 and later) is minimal (see for example Figure S4). [114-118]

8. P-CaMKII staining --- one of the most beautiful P-CaMKII staining I have seen. However, it requires control of antibody in side-by-side control with either CaMKII KO or ideally CaMKII T286A mice.

Unluckily, we had neither CaMKII KO nor CaMKII T286A transgenic mice available. From the non-stimulated controls that we included, it appears licit to say that the antibody preferentially recognizes the phosphorylated form by looking at the integrated signal intensity along dendrites from different samples. The antibody mAb 22B1 is a very well validated anti P-CaMKII antibody, has been extensively used in literature, and has been successfully employed in immunostaining experiments (Ouyang et al, J.Neurosc, 19(18), 7823–783, 1999). Importantly, it has been shown to be dependent on phosphatase inhibitors (Kindler and Kennedy, J. Neurosc. Methods, 68(61-70), 1996), therefore we included them in every step of our IF procedures (see Materials and Methods) and was tested on T286A mice extracts (Cooke et al, J.Physiol, 574.3, 805-818, 2006). We are therefore confident that the signal highlighted by this antibody is due to P-CaMKII. [418-420]

Minor points:

Line 36: “aninput” should be “an input”

Corrected in text. [45]

Fig. S3A +BDNF Images: they contain irregular gray rectangles.

For display purposes, we straightened the depicted dendrite by joining the straight tracts of the dendrite in a linear way, in analogy to others (for example, Dieterich et al, Nat Neurosc, 13, 897-905, 2010). Therefore, some regions of the final image presented in Figure S3 outside the dendrite could not be reconstructed due to the curvature of the dendrite. We therefore identified those background regions with gray rectangles. An explicit description has been added to the Figure legend. [S31-S32]

Fig. S3A +Saline image: the order of images appears to be wrong.

We amended the image accordingly.

Fig. S7A: it appears that there are two gray crosses, but not explained in the legends what they are.

Gray crosses represent the average (EI; CI) for palmitoyl-Cherry and Arc-palmitoyl Cherry. The Figure has been removed because it was referred to one-photon GCaMP excitation.

Reviewer #2 (Remarks to the Author):

Recommendation: This paper has some nice ideas, but in its current form is too messy to publish. It was actually quite hard to review. The reasoning for making ArcSYP-ChR2-Cherry in the first place is not even clear? How would such a tool be used? The use of the word “engram” implies in vivo use, but I can’t see how such a tool would be useful in vivo. It might be more spine-enriched than cytoplasmic ChR2, but the massive soma expression is going to lead to neural spiking and back-propagating action potentials, which will negate the supposed spine-specificity of the manipulation. Is this a slice tool? Cultured neurons? The in vivo portion of the Discussion is unconvincing.

We thank Reviewer #2 for the criticisms and comments; we did our best in trying to answer the mentioned issues. We accept the criticisms by Reviewer #2 that the in vivo ambition of the paper (linked to the engram objective) is not directly experimentally addressed in this manuscript, and therefore we have eliminated much part of this description from both Introduction and Discussion, leaving some perspective arguments about possible future in vivo applications at the end of the manuscript. The revised paper is now more factual, and linked directly to the description of the experiments presented and to their direct implications. In any case, we have added new data in vivo, adding credibility to the future in vivo applications, also in light of the remarkable lack of somatic expression in vivo, and synaptic selectivity (Figure 5), as described and discussed below. Accordingly, the sections in the Discussion that refers to in vivo applications are functional to the new data provided.

Major things:

- A lot of statements are unnecessarily made too strong. For instance, “No causal role has yet been demonstrated for the modifications of synapses in the formation of memories.” Really, a great number of people would be surprised to hear this claim. Of course synaptic modifications have been shown to be critically involved in memory formation, strengthening, recall, etc. But there are many details yet to be worked out...

We followed Reviewer #2's indications and modified the Introduction and Discussion sections accordingly. In particular, the indicated statements have been removed and the whole sections reformulated. We are of course aware of the precious work of many excellent researchers investigating the role of synapse potentiation in memory formation. Our aim was not to criticize their work, rather to highlight the ongoing effort on strengthening the evidence for the link between the two aspects, which still remains an area of active research of great interest in the community (see for example Neves et al, Nat Rev Neurosc, 9, 2008; Rogerson et al, Nat Rev Neurosc, 15, 2014 or Poo et al, BMC Biology, 14, 2016).

However, the text has been thoroughly revisited with a more balanced account of the literature, following the indication of Reviewer #2. [22-27; 32-39]

- “Indeed, the current spatial resolution of opsin expression is limited to whole cells but does not allow selective subcellular localization control.” This statement is patently false. Opsins can and have been targeted to the axon initial segment, to the soma, to the mitochondrion, to dendrites, to synaptic vesicles, etc. Some of this work has been done a decade ago. I really have no idea what the authors mean by this. The statement could not be more incorrect. The authors then go on to cite some of these papers but say that they lack sufficient specificity.. ???

We understand that our message in the Introduction was likely misunderstood. We are aware of the trafficking motifs that have been added to various opsins to influence their localization, and we did not mean to underestimate their value but only to underlie the difference with our approach. That body of published work is focused on the subcellular trafficking to various subcellular parts of the neuron. What we meant, and what we aim at demonstrating in this work, is to develop a method for the activity-dependent expression of opsins at synapses, to genetically tag active neurons (that usually employs the c-fos promoter, as in the fos-trap case) at a synaptic level, a method that is different from those approaches and that is currently missing.

We modified the Introduction accordingly and clarified the issue. [32-39]

- the authors should cite some of the papers on “single-synapse optogenetics.” Some of those have been very useful. We made sure that relevant papers that introduce single-synapse optogenetics approaches are cited in the text. We may include in this list References 8, 22, 50, 87.

- I think the data provided by the authors do not support the claim “Arc RNA sequences determine the uneven tagging of synapses, while the SYP tag docks the protein inside the synapse.” The data in the paper are kind of all over the place on those points.

We reorganized Figure 2 and the accompanied description in the text to address the comments of Reviewer #2 on this topic. Arc sequences control the expression of SA-Ch at synapses, whereas the SYP tag (in the revised nomenclature, the SYN-tag) confers PSD localization. We dissected the role of the two parts in the final SA-Ch construct. In the top panel, we show that the presence of the variant bearing the SYN tag preferentially localizes in correspondence of the PSD, whereas in the lower panel we compare SA-Ch and S-Ch, which have the identical coding sequence but differ from the presence of Arc UTR RNA sequences. Whereas S-Ch tags the majority of spines (as also shown in the comparative data in Figure 1d) irrespectively of their activation status, SA-Ch tags active spines only

(characterized by concomitant SEP-GluA1 exposure). We conclude that Arc sequences are instructive on whether a spine expresses SA-Ch or not. In addition, S-Ch enrichment to spine is very similar across different levels of SA-GluA1 enrichment. [119-124; 132-139]

However, we understand that the term “uneven” might sound obscure, so we eliminated it from the text. We thank Reviewer #2 for prompting us to clarify the issue.

- the images that contain a cell body make it clear that the vast majority of the ChETA, with all the targeting motifs, remains in the soma. The authors make such a big deal about needing it not to be expressed there that it's surprising that they don't comment on this.

We agree with Referee #2 that the ratio of synaptic versus somatic expression is a crucial parameter to be characterized. First of all, the new in vivo data show that the ratio of synaptic versus somatic expression in the in vivo hippocampus is very favourable to the synaptic expression; our in vivo expression experiments, following in utero electroporation of the SA-Ch plasmid, show that, in the in vivo hippocampus, SA-Ch is largely excluded from somas, while expressed at synapses along in basal and apical dendrites. This is qualitatively shown in the images in Figure 5 and quantified in Figure S7, by plotting Cherry intensity profile of CA1 neurons; we took advantage of the anatomical organization of CA1 pyramidal neurons, where somas are densely packed in the stratum pyramidale and dendrites are mostly confined in the upper and lower layers. Thus, in vivo, we find that Cherry fluorescence is most prominent in the dendritic layers. This finding settles in a convincing way the issue of the somatic versus synaptic expression in vivo. [209-220; S71-S77]

As for the in vitro situation, to which the Reviewer's comment refers, we regard the residual somatic expression of SA-Ch observed in dissociated cultures as a phenomenon due to overexpression: i) the promoter used in the in vivo experiment is a weaker one than that used in the in vitro experiment and ii) as the dendritic arborization of neurons grown in culture is significantly smaller than the one of in vivo neurons, and therefore an excess of transcribed RNA may prevent it from being further accommodated in dendrites in a repressed state, due to a saturation of the dendritic transport and associated translational repression system. Under in vitro overexpression conditions, it is therefore more likely for some RNA molecules to escape the molecular machinery of the neuron that prevents them from being translated. In addition, somatic expression in cultured neurons was most evident in stimulated neurons, with respect to unstimulated conditions; however, stimulations performed in cultured neurons are a global treatment that can signal the transcript to be derepressed also in the soma. [114-118]

Thus, as already written above and in the reply to Reviewer #1, under in vivo conditions the degree of somatic activation compared to that in spines is expected to be very favourable, also because the stimulation that CA1 neurons received was a more natural one. [210-214]

- one thing that's very confusing is that the authors don't do a good job distinguishing between the fact that ChETA-mCherry (even with the targeting motifs) seems to preferentially accumulate in large spines over small ones at baseline, and then the fact that GCaMP6 signal is stronger from the large spines. Maybe they underwent LTP, or maybe they just expressed more ChETA and/or GCaMP6. The correspondence between spine size and LTP/synaptic strength is incompletely understood.

We revised the manuscript according to the Reviewer's comments. As also suggested by Reviewer #1 and #3, we modified the setup for SA-Ch activation by imaging GCaMP6 with 2-photon stimulation, and expressed the signal in $\Delta F/F$ units, that are relatively independent of volume and concentration. This new experiment appears in Figure 4a. [167-172; 397-410]

Also, we quantified the dependence of SA-Ch enrichment on PSD size using Homer1c-EGFP, a good marker of spine volume (Meyer et al. Neuron 82, 430–443, 2014). In addition, we show that SA-Ch expressing spines are also exposing SEP-GluA1, that is exocytosed to the surface of spines following potentiation (Patterson et al, PNAS, 107, 15951-15956). We thank Reviewer #2 for prompting us to clarify the issue. [128-139]

- far too many details are left out for even an expert to reconstruct how the experiments and data analysis were done. What were the lookup tables for GCaMP signal binning? Those images look too clean as it is right now- I think the lookup tables have been adjusted to show the desired outcome. What laser powers were used? What excitation and emission filters?

Detailed experimental data have now been included in the revised text, in the Materials and Methods section. We repeated the experiment by imaging GCaMP6 by setting the Ti:Sapphire laser at 990 nm with RM690 filter. Actual peak was detected at 988 ± 2 nm. Channelrhodopsin was excited with 10ms 488 nm illumination with the same filter setup. Under these conditions, the measured power was 8.9-10.7 μ W. Data are expressed in $\Delta F/F$ units after dark frame

subtraction. For dark frame calculation, identical acquisitions were performed with all shutters closed, thus no laser passing. [167-172; 397-410]

- the writing is rough throughout. I had to reread some sections 3-4 times to figure out what was going on.

We re-wrote the text trying to make it clearer.

Medium things:

- In Fig. 1Aii-ii', the authors say that mCherry was quite uniform along dendrites, when instead it is obviously blebby, as typically happens with dendritic expression of mCherry.

That statement was dropped from the revised text, as our focus was on the change in expression in dendrites following KCl treatment. [58-64]

- the authors say that "SYP-Ch was evidently enriched at spines compared to unmodified ChETA-Cherry", but that doesn't seem clear from the images.

We tried different lookup tables to try to make it visually clearer. Enrichment is quantified in Figure 1c; we added an explicit cross reference to it in the text, to further support our statement. [95]

- also, they say that large synapses preferentially express Arc-Ch over smaller ones, but there really is no quantification. The images are pretty bad, too, which doesn't help interpretation.

We quantified this dependence comparing SA-Ch and Homer1c expression in Figure 2e, a good indicator of spine volume. [128-132]

- the authors say that the Arc DTE is "the best candidate" for localization, but then in Figure S1 they show that CaMKII and MAP2 are 3 times better than Arc. Very confusing.

We apologize if Figure S1 was misleading. Arc DTE was chosen as the best candidate because it displayed a lower dendritic expression under non stimulated condition; however, KCl strongly induced its dendritic expression, giving the best dendrite-to-axon ratio after stimulation. CaMKII DTE also increased dendritic expression upon stimulation, but to a much lesser extent. [54-64; S2-S5]

- Figure S4 fairly clearly shows that after KCl treatment, both ArcSYP-ChR2 mRNA and protein are preferentially localized to the soma. The whole point of the paper is that this should localize to spine heads.

We already addressed above the issue of somatic expression of ArcSYP-ChR2 protein in the response to the 5th previous major point of Reviewer #2 (as well as of Reviewer # 1). We only stress once again that the ArcSYP-ChR2 protein is indeed expressed in activated spine heads, particularly so in the in vivo hippocampus.

As for the ArcSYP-ChR2 mRNA, the mRNA imaging system (MS2) we employed tags RNA both in translationally-repressed and translation-competent states; thus, MS2 protein recognizes also repressed RNA that travels through the soma before reaching dendrites. Constitutively expressed EGFP-MS2 contains a NLS sequence (as explicitly stated in Methods) and excess EGFP-MS2 protein is localized in the nucleus, which explains most of the signal observed in the soma (please see for example the first cell in the image, which only express EGFP-MS2). The use of NLS in MS2 constructs is very typical in the literature (Fusco et al, Curr Biol, 13, 161-167, 2003; Park et al, Science, 343, 422-424, 2014). [333; S43-S44]

In any case, what matters for our purpose is the activity-dependent localization of ArcSYP-ChR2 protein to the spines. The other information is for a complete and thorough description of the experimental system.

- the flux through ChR2 is probably a negligible source of spine $[Ca^{2+}]$. Depolarization of the spine head will open voltage-gated Ca^{2+} channels, that are going to be the bulk of the source of spine $[Ca^{2+}]$.

We thank the Reviewer for this suggestion. We agree with the Reviewer's comment and suggestion, and we included new experiments with voltage-gated calcium channels inhibitors (nifedipine, Ni^{2+} and Zn^{2+}), as reported in Figure 4a. These drugs/treatments inhibited the light-dependent calcium influx. Importantly, the opening of such channels was dependent on light stimulation of Channelrhodopsin. The data is presented in the new Figure 4, and described in the

text. [173-175]

- in Figure 4, it's not clear how the Chr2 is being activated. It is being activated by the GCaMP imaging light? Or is a separate light source illuminating it?

In the experiment the Reviewer refers to, in the original manuscript that was reviewed, Chr2 was activated by the same GCaMP imaging light. We changed the experimental setup following the Reviewer's suggestions, presenting the new experimental data in Figure 4. [397-408]

- In Fig. 4A, only 2 of the panels show the soma, so it's impossible to tell how well the targeting worked. The authors say, "The expression pattern of Arc-palmitoylCherry was very similar to ArcSYP-Ch (and Arc-Ch) (Figure 4A)." I thought their whole claim was that ArcSYP targeted things to spines. And instead of "Arc-Ch", do they mean "SYP-Ch"?

With "expression pattern" we meant sites of expression associated with spines, i.e. where translation took place. In the original version of Figure 4 the Reviewer refers to, both Arc-palmitoyl Cherry and ArcSYP-Ch neurons presented sparse labelled spots in dendrites, in contrast to S-Ch and palmitoylCherry that labeled dendrites with continuity. SYP tag is responsible for the enrichment of the protein moving from the initial site of translation to the spine head. By saying that Arc-SYP is similar to Arc-palmitoylCherry and ArcSYP-Ch, we meant that they all display a spot-like pattern along dendrites (please see Figure 1b) that is due to local translation.

Note that this sentence was referred to a version of the Figure that is no longer present in the manuscript, so it was deleted from the revised text.

- the spine GCaMP6 signals are harder to interpret than the authors let on (or perhaps realize). From lots of experience using it, smaller spines tend to show higher DF/F than larger ones. The reason almost certainly is that as Ca²⁺ comes into a much smaller volume, the local concentration increases picked up by GCaMP are much higher. And since the Hill coefficient of GCaMP6 is >3, this translates into a much greater photon flux from fewer highly-saturated GCaMP6 molecules than from more less-saturated GCaMP6 molecules. It seems that they're largely comparing the ratio of GCaMP signal at spines to shafts, but they might need to get into a discussion of how absolute GCaMP signal is not reflective of spine strength.

We modified the experiment as asked by the Reviewer's comment (see Figure 4) and have modified the description and discussion of the experiment accordingly. [166-175]

- what is the source of the GCaMP6 signal in the cells in TTX and not expressing Chr2?

TTX was only present in some of the experiments regarding Chr2. In the previous version of Figure 4B, neurons expressing SA-Ch that were imaged in presence of TTX are reported the second graph on the right, whereas all other samples were recorded in standard, TTX-free, ACSF. In the two-photon experiment reported in the revised version of Figure 4, SA-Ch samples are imaged in standard ACSF. Light-dependence is confirmed by the temporal profile of the registered signal as well as by its absence when the 488 nm stimulation was excluded, as reported in the new Figure 4a. Three samples are included, depending on the imaging medium: (i) ACSF, (ii) ACSF with Voltage-gated calcium channels inhibitors, and (iii) TTX. Their presence is explicitly indicated in the revised text as well as in the new Figure 4a. [171-172; 397-401; 826-829]

- when the authors say, "even single inputs could drive significant localized inputs that were not overcome by the activation of the whole rest of the neuron (Figure 4D)", then doesn't that throw into question the whole point of trying to focus on spine-targeted Chr2?

We think that the effect on the neuron will be dependent on the number, distance and relative position of excited spines, as well as the pattern of the stimulus, as it happens with physiological neurotransmission. In any case, since the sentence was referred to an experiment no longer present in the text, it was removed from the revised version.

Minor things:

- the authors don't define G418 as geneticin.

Corrected in the text. [108]

- the labels seem to be swapped for Fig. 2D.

Correct in the manuscript.

- line 525: “vi” should be “iv”

No longer present in the revised manuscript.

Reviewer #3 (Remarks to the Author):

The manuscript by Gobbo et al., titled: “Activity-dependent expression of Channelrhodopsin in neuronal synapses” describes a novel targeting strategy for optogenetic tools, designed to express a light-gated ion channel specifically in previously-activated synapses. To target the opsin into the synapses, the authors utilized a clever duplex strategy combining RNA- and protein-targeting sequences that achieve very high specificity to post-synaptic sites and demonstrate robust activity-dependence. Addition of the immediate-early gene promoter Arc modulates translation of the so targeted mRNA in an activity-dependent manner. The imaging and immunostaining data convincingly demonstrate synapses-specific targeting as well activity-dependent expression. The authors then use calcium imaging to evaluate the efficiency of stimulation of ChETA-expressing synapses with light, quantifying GCaMP6 fluorescence and phosphorylated CaMKII as proxies for induced synaptic currents.

Overall, this article has tremendous innovative potential and I am certain this work will be interesting for many neuroscientists. I believe that with a few additional experimental controls as detailed below, this manuscript would be highly suitable for publication in Nature Communications. I would be more enthusiastic if the authors find a more convincing way to quantify the efficiency of reactivation of synapses with light. Inclusion of electrophysiology experiments will allow a more precise quantification of light-evoked currents and comparison with real synaptic currents in the same neurons. Some additional experimental data is missing, and its inclusion would facilitate reproduction of these experiments by other labs.

We thank Reviewer #3 for appreciating the ideas we put forward, and for the constructive comments. We did our best in trying to answer the mentioned issues, and we think that thanks to Reviewers’ suggestions our work has significantly improved.

Specific comments:

1. My major concern is that it’s unclear whether GCaMP6s fluorescence changes are due to calcium influx through ChETA. This variant does not have an especially high calcium conductance and is engineered for short opening time and therefore conducts in absolute terms the least amount of ions compared to other ChRs. The paper cited in reference 41 actually doesn’t contain any data relating to ChR2 as calcium-permeable. Reference 42 demonstrates a detectable ChR calcium influx at 80mM extracellular calcium. This is very common in the community (for a good reference, see Schneider et al., Biophysical Journal 2013), but I am not aware of any published data set in which such strong calcium-influx through ChR was detected at physiological conditions. I argue that the effect the authors see is due to ChR induced voltage-deflection which partially open high conductance voltage-gated calcium-channels. Therefore, an experiment with nifedipine, barium or cadmium would answer this question.

We thank the Reviewer for the suggestion regarding the possible source of calcium influx upon light stimulation. To investigate this in detail, we included in our revised GCaMP6 experiments treatments with VGCCs inhibitors, as proposed by the Reviewer. We employed a combination of nifedipine, Ni²⁺ and Zn²⁺ to block all types of VGCCs. VGCCs inhibitors drastically reduced calcium influx following light stimulation. Importantly, the opening of such channels was dependent on light stimulation of Channelrhodopsin. The data is presented in the new Figure 4, and described in the text. [173-175; 826-829]

2. The experiment in Fig 4 A are difficult to interpret since ChETA and GCaMP are excited in parallel – so a stable baseline is missing. A cleaner approach would be to use red-shifted genetically encoded calcium-indicator such as jRCaMP or a red-shifted chemical dye. Again I am very doubtful that the transient influx upon illumination is really due to higher conductance of the O1 state. Since the authors seem to be able to use a two-photon microscope, combining two-photon calcium imaging with full-field light stimulation might also allow a stable calcium baseline.

We need to thank again Reviewer #3 for the valuable suggestions. Unluckily, the use of red (or orange) calcium indicators would have forced us to change the fluorescent protein that is integral part of SA-Ch. Instead, we imaged GCaMP6 with a two-photon laser tuned at 990 nm, while a short-pulse stimulation of the Channelrhodopsin was performed at 488 nm. We recorded the GCaMP6 signal in the dendrite and at the base of the spine, and the stimulation was performed on the spine head. This allowed us to express the data in $\Delta F/F$ units. [168-172; 397-410; 826-830]

3. The targeting sequence utilizes an NMDA consensus sequence as well as a PSD95 binding site. Utilizing these sequences might alter intrinsic electrophysiological properties of the synapse by competing with endogenous binding partners of PSD95. A rough estimate of the number of targeted ChRs ($35\text{pS}/\mu\text{m}^2$ Grossmann, J. Computational Neuroscience 2013 or Zimmermann et al, 2006, Biochemical and Biophysical Research Communication), and an average PSD95 area of $.02 - 0.3 \mu\text{m}^2$, the expected voltage deflection will be rather small. I still believe that it is possible to mimic postsynaptic potentials, I would just urge the authors to provide electrophysiological data which will more clearly demonstrate what kind of responses are induced as well as whether endogenous synaptic properties are changed. These experiments will provide the necessary strength which will make this work more convincing for a lot of readers.

The new GCaMP experiments that we did following the Reviewers' indication (Figure 4) record calcium influxes upon 488 nm stimulation in spiral scanning mode, so that we could focus the illumination on the desired region. Thanks to the physical properties of the light path and the sparsity of SA-Ch spines that we analyzed, we are confident that these come from a single spine stimulation. [407-408]

Concerning the point raised by the Reviewer regarding competition with endogenous partners (in particular, NMDAR), we quantified the surface pool of AMPA and NMDA receptors in neurons expressing SA-Ch vs. control neurons, and the relative data are integrated in Figure S2d. We found no difference in the ratio of the sAMPA/sNMDAR signal in dendrites from the two populations. We considered the ratio of the two signals because the absolute value is dependent on the number of synapses and is rather variable across different neurons. Instead, sAMPA/sNMDAR ratio is an independent measure of the relative presence of the two receptors at the synapses. Indeed, neurons tend to co-regulate their expression in order to maintain a constant AMPA/NMDA current ratio (Watt et al, Neuron, 26, 659-670, 2000). [79-80; S19-S23]

Nevertheless, we hold the Reviewer's comment in high consideration, and we will make sure to include electrophysiological data as next step of our work.

4. In figure 1C, it seems that Arc enrichment is not significantly different from with BDNF, but for the spine count, measured as "fraction of tagged spines", the BDNF treatment increases the number of spines drastically. Does this mean that Arc only increases expression in previously non-tagged spines and is not actually increasing the expression level of already expressing spines?

For the way the parameter is calculated, we could not exclude this possibility. However, we think that BDNF could increase expression also at spines that already express A-Ch without necessarily increasing the EI value. In fact, EI compares Cherry intensity at the spine with the intensity in the dendrite 1-2 μm away from the spine. After translation, A-Ch can diffuse both into the spine and on the near dendrite. In this case, as the intensity would increase in both regions, the EI parameter would be affected marginally. Indeed, in SA-Ch samples, whose translation should be regulated in the same way, BDNF also increases the EI value. [112-114]

5. From the figures, it appears that ChRs are also expressed to substantial amounts at the neuronal soma. This expression should be measured and quantified in comparison with non-targeted channelrhodopsins. Sufficient functional expression of ChR in the soma could explain the c-fos data. Furthermore, under more natural conditions, in which spikes can be readily generated at the soma and back-propagate to the dendrites, the effect of the activation of specifically targeted ChRs in the spines might be overwhelmed by the somatic spiking driven by this non-specific expression. How do the authors intend to overcome this limitation when this tool is applied in vivo? Perhaps an addition of a somatic destabilization sequence would bias the turnover rate of somatic channels?

We agree with Referee #3 that somatic expression could be a limitation, as observed under the in vitro conditions, as also discussed extensively above in this document. However, our new in vivo experiments show that in the hippocampus SA-Ch is largely excluded from somas, while expressed at synapses along in basal and apical dendrites. This is qualitatively shown in the images in Fig 5 and quantified in Figure S7, by plotting Cherry intensity profile of CA1 neurons; we took advantage of the anatomical physical organization of CA1 pyramidal neurons, where somas are densely packed in the stratum pyramidale and dendrites are mostly confined in the upper and lower layers. Thus, in vivo,

we find that Cherry fluorescence is most prominent in the dendritic layers. We therefore regard the residual somatic expression of SA-Ch observed in dissociated cultures as a phenomenon due to overexpression: (i) the promoter used in the in vivo experiment is a weaker one than that used in the in vitro experiment and (ii) in vivo the dendritic arborization is much larger than in culture, where an excess of transcribed RNA may prevent it from being further transported in dendrites in a repressed state, due to a saturation of the dendritic transport and associated translational repression system. Under in vitro overexpression conditions, it is therefore more likely for some RNA molecules to escape the molecular machinery of the neuron that prevents them from being translated. In addition, somatic translation in culture was most evident in stimulated neurons, with respect to unstimulated conditions; however, as Reviewer #1 noted, stimulations performed in cultured neurons are a global treatment that can signal the transcript to be derepressed also in the soma. Thus, under in vivo conditions the degree of somatic activation compared to that in spines is expected to be minimal. [114-118; 209-220; S71-S77]

Even if that wasn't the case, the anatomy of dendrites in the hippocampus could be used to avoid soma excitation. There have been some very promising hardware strategies to shape the illumination field. For example, Ferruccio et al. (Neuron, 82, 1245–1254, 2014) developed a nice light guide to control light emission points along the z-axis. The use of such a device could be employed to selectively illuminate the dendritic layer minimizing the amount of light delivered to the pyramidal layer.

6. Fig 5 C is a very convincing experiment – the mCherry negative spines (empty boxes) are analyzed from transfected neurons. Does this imply that only spines which had neighboring mCherry positive spines were taken into account in the analysis?

Indeed, mCherry negative spines are analyzed from transfected neurons only. We took images of identical dimension and considered all spines from transfected neurons, then divided them into the Cherry-positive and Cherry-negative categories. Thus, all negative spines from such dendrites were taken into account, not only those adjacent to Cherry-positive spines. Indeed, some of these negative spines had no Cherry-positive immediate neighbour, but belonged to a dendrite having Cherry-positive spines. [187-191]

Minor points:

Line 244: Please specify which sequences you tested and add them in the supplements

We included necessary information in the Supplements. [S86-S134]

Line 44: What is this minimal sequence – please present it in the supplements. The reference to citation number 20 is confusing since it implies that the minimal sequence of Arc is published there. I could not find this information in this reference.

We apologize for the confusion. Indeed, the Authors of the cited reference test a certainly high number of different parts of the Arc transcript. Minimal Arc DTE encompasses nucleotides 2162-2513 of the authors' sequence (as reported in Figure 3 in Kobayashi et al, Eur J Neurosc, 22, 2977-2984, 2005). From their work it emerges that different parts of Arc 3'UTR have dendritic-targeting activity; however, Arc minimal DTE 2162-2513 is responsible for the labeling of most distal dendritic portions (Figure 2H and 3D in the paper). They suggested us to also clone flanking sequences to make sure that the RNA context would not interfere with its folding (H. Kobayashi, Osaka University, personal communication), so we cloned nucleotides 2035-2702 from plasmid pCMV-ArcF cited in the paper. We included the sequence of the DTEs used in the Supplementary Information. [332-333; S88-S99]

Line 19: Channelrhodopsin is an opsin, but not all opsins are ion channels.

We corrected the text accordingly. [28]

Line 20: The Malinow paper specifically modulates synaptic inputs to the amygdala – it is given as ref 7 in your manuscript. You might want to rephrase to “modulating specific synapses”. The same applies to the statement in line 220 – there are in fact a few papers describing synapse-specific manipulations. We

agree with Reviewer #3 and we rephrased the text accordingly. [29; 245-251]

Line 36: space missing between “aninput”

Corrected in the revised text. [45]

Line 42: Please provide the reference for the “fast maturing mCherry”.

We added the reference as requested. mCherry is reported to fold in about 15 minutes at 37°C. [53]

Figure legend S1: Please change “I is ...” to something like “Intensity (I)” – it is a bit confusing.

We changed the legend as requested in the revised text. [S7]

Figure legend S1: Please provide further information regarding how many cells and how many preparations are included in the data set. Similar to the information you provide in the spine data set in the Statistics part of the manuscript.

We indicated the numbers of dendrites and cells included in the data set in the new version of Figure S1. [S6]

Line 72: “Conversely, SYP-Ch distribution was quite similar to the EGFP one under the same conditions.” This sentence is confusing, since it is not really the same conditions. isn't it better to call it (-) BDNF?

We apologize if the text was misleading. We compared the difference between A-Ch and S-Ch (with respect to the EGFP channel) following BDNF treatment. A-Ch transcript has dendritic localization sequences in the UTR, whereas S-Ch does not have localization sequences. Our data indicate that BDNF stimulates dendritic translation of A-Ch (as it significantly deviates from the EGFP signal, which is present in the dendrites as a consequence of diffusion from the soma). The behaviour of S-Ch, on the contrary, after BDNF stimulation, is much closer to EGFP, indicating somatic translation as its source. We modified the text and the figure S3 to state it more clearly. [83-87]

Line 85: Is the comparison depicted in Figure 2A significant?

Yes, it is. We apologize, significance indication went missing during the preparation of the figure. This has been corrected in the revised version. [798]

Line 92: “smaller spines were not as effectively labelled as the dendrite (Figure 1B)” – this statement is meaningless without some quantification of the imaging data.

We agree with Reviewer #3 and we dropped the line from the revised text. We did not quantitatively correlate ChETA expression and spine size. As shown in Figure 1c, neurons expressing unmodified ChETA have EI close to 1, which means that, statistically, some spines have a EI<1, therefore they are not as effectively labeled as the dendrite. [102]

Line 126 – Could it be that the expression of ArcSyp-Ch is actually increasing size of synapses? Does Arc-Syp-Ch and Arc-Ch constructs target different sized synapses after excitation with one of the chemical LTP protocols (Figure 1 D). This might indicate that Arc-Syp-Ch actually causes the increase in synapse size.

Although not quantitatively measured, we did not observe any evident difference in spine size between samples expressing different constructs that underwent the same treatment. Also, spines from SA-Ch neurons look normal and show no difference than control neurons (Figure S2). [78-79]

Line 134 – Why did you include Forskolin in the extracellular medium? Is forskolin required for activity-dependent expression?

The presence of Forskolin in the medium is recommended by the LTP protocol we used, described in the reference paper (Hill and Zito, J.Neurosc, 33, 678–686, 2013). Furthermore, forskolin (or an analogue increase in cAMP concentration) is required for stable LTP expression (see for example Barad et al, PNAS, 95, 15020–15025, 1998, or Otmakhov, J Neurophysiol 91, 1955–1962, 2004). Gobert et al. (J Neurochem. 2008, 106(3), 1160–1174) show that forskolin increases 5'TOP RNAs, which include dendritically localized mRNAs. Therefore forskolin is required for activity-dependent expression under these experimental conditions.

Fig 3 – The switch in terminology from Arc-Syp-Ch to AS-Ch is confusing – it occurs a few times in the different

figures. Also, “pre” and “post” in the context of synapses is misleading – maybe better use “before” and “after” as used in the text

We thank Referee #3 for the useful comments on the nomenclature. We now uniformed it throughout the text and the figures, as it was anticipated in our letter. [75-77]

Also regarding nomenclature: the full construct is “N-ChETA-mCherry-SYP-C”, and in this case I find it a bit confusing to abbreviate it as SYP-Ch – please consider revising the nomenclature.

We changed the name to S-Ch; we think that this may help reducing confusion. We hope the new nomenclature may be sufficient in readily identify the modifications of the various constructs. We hope that this notation will be useful when applied to different proteins as prefix to the protein of interest. [76]

Based on the schematic drawing in Figure 5 A and S6 – the experiments are performed less than a day after transfection. How does the expression pattern look after 2 – 3 days? Are there more synapses labeled? Do the authors see a similar increase of tagged synapses as shown in figure 1D? This would be highly important for in vivo studies.

In our in vitro experiments, we maintained the time course of expression as fixed as possible, to be sure that differences in expression were not due to the time of expression. Indeed, as we underlined above, we avoided neurons to overexpress the transcript from constitutive CMV promoter for too long. We also reasoned that, for an effective encoding of a specific context/stimulation, the temporal expression of SA-Ch transcript should match as closely as possible the time of the exposure of the stimulus. Accordingly, in our in vivo experiments, we controlled transgene expression with doxycycline, using the TetON system. When doxycycline is administered intraperitoneally, expression starts to be detectable shortly after (Zhu et al, PlosONE, 6, 2007). We agree with the Reviewer that, in vivo, timing is a crucial factor; we therefore evaluated the expression after 2 days, and after 3.5 days from the start of the treatment, which was repeated daily. Prolonging the time of transcript expression may increase the number of labeled synapses up to a certain extent in absence of novel stimulations (compare for example Figure 5a and Figure S7c, where representative images of mice exposed respectively 2 and 3.5 days to doxycycline are presented). Rather, we found that the exposure of the animal to a novel environment significantly increases the number of labeled synapses (Figure 5). [217-220; 433-436; 851-856; S71-S78]

Can the authors provide the exact acquisition parameters for the calcium imaging –light intensity, frame resolution – it is a difficult experiment! Was frame size always the same?

We included all experimental data regarding the new version of the experiment. They are summarized in the main text in the Results section, and reported in detail in the Materials and Methods section. We maintained the frame size and dimension constant, giving an interval of 20ms between two consecutive frames. [397-410]

Figure 3 – to evaluate the size of synapses the authors use the mCherry fluorescence – I don’t think it is that straight forward to transfer the theory of a diffusible filler of Yasuda (ref 37) to a membrane protein. Especially when the construct is supposedly anchored to PSD95. Do you actually see an increase in size of the fluorescence or is it only intensity that increases?

We completely agree with the Reviewer. We thank the Reviewer for noticing, and we modified the text of the fourth section of the Results to clear the point. To answer the comment, we measured the volume change by measuring the soluble filler EGFP intensity in the spine, confirming volume change of the stimulated spines (new version of Figure 2). In cultures, we noticed that expressing spines tend to be larger by looking at the EGFP channel; to quantitatively measure this, we correlated SA-Ch expression with Homer 1c signal, a well-validated indicator of spine volume. [128-132]

In the focal LTP experiments described in Figure 3, two mechanisms could be responsible for the observed increase in Cherry fluorescence: (i) increased accumulation due to volume (and PSD size) change, and (ii) a contribution from local translation. S-Ch served as a control to evaluate the contribution of (i); accordingly, we reasoned that the remaining part of the observed increase can be ascribed to local translation (point ii). Indeed, anisomycin blocked this increase in fluorescence intensity (new Figure 2); however, although useful, anisomycin treatment could not help us estimate the contribution of volume change alone, as inhibition of translation also impairs the long-lasting volume increase (please see Figure 2c2). [151-155]

What are the absolute expression levels for Arc-Ch and Arc-Syp-Ch? If Arc-Ch is significantly lower or higher expressed, a non-linear response curve for the photodiode could lead to the difference of relative fluorescence ratios.

Although we did not quantify the absolute levels of SA-Ch and A-Ch, images were acquired with the same laser power and PMT gain setup. Thus, since we did not see any striking variation in intensity using the same imaging parameters, our overall conclusion is that the expression level of the two-constructs is by any means comparable.

A membrane-bound Arc-Syp-Palm-mCherry construct as could have been a better control to see whether expression of ChR is actually changing size of synapses.

We didn't see any evident increase in spine size when considering the whole spine population, in particular between SA-Ch and S-Ch expressing neurons. If the SYN tag is responsible for the increase in spine size, we would expect to see a global increase in spine size in S-Ch (+) neurons, as it is actually present in most of the spines. Furthermore, when we focally stimulated single spines, volume change temporally precedes SA-Ch expression (please see Fig. 2c1,d1), so we think that it is unlikely that expression of ChR is actually changing size of synapses. [77-79; 820-823]

In any case, the panel we think the Reviewer refers to is no longer present in the revised manuscript.

Figure S3 – the scale bars in the images are 10µm while the profile plot is 200µm – but the image is not 200µm large. I found it confusing since it indicates that this is the part of a larger image which was used to analyze and create the graph.

We checked the images as suggest by Reviewer #3. The graph used to the dendrite depicted in the (+) BDNF panel just above the plot profile as sample, and is actually 200µm long. We modified the image accordingly to remove any confusion.

Figure 1D – the title “spine count” is confusing – maybe label axis “fraction of tagged spines”?

We thank Reviewer #3 for the advice and modified the axis label as suggested. [786]

Reviewers' comments:

Reviewer #1 (Remarks to the Author):

The authors have addressed most concerns from me and, apparently, from other reviewers. Overall the manuscript is improved significantly. I recommend accepting the manuscript.

Reviewer #2 (Remarks to the Author):

Recommendation: This revision of the manuscript is much improved. It's still not 100% clear to me how useful this reagent is going to be under difficult in vivo conditions, but it's a reasonable advance.

Major things:

1. The manuscript is still very hard to follow (but much improved). It seems like some presentations of data are not consistent. For instance, Fig. 1c-SA-KCl-SA-Ch and Fig. 1d-SA-Ch show that KCl increases the enrichment index and positive spine fraction, but Supp. Fig. 4a would seem to indicate precisely the opposite. Another example is that the images in Fig. 2d show more punctate labeling for A-Ch than for SA-Ch, and this is presumably one of the best images obtained. Another one is that Fig. 4c shows that as much SA-Ch accumulates in spines in the dark as the light.
2. All bar plots should be presented as box plots, or even better, showing all data points like in Fig. 4a (why was this the only one to do so?). It's just too hard to tell the reliability of the statistical tests, particularly since the authors are showing standard error instead of standard deviation (the latter would be much better).
3. The in vivo data in Fig. 5 isn't super-convincing. It seems like they have to go to some lengths to find differences with the spine-neighbor calculations. Again, better presentation of the bar plot data would help evaluate these things.

Lesser things:

1. The imaging settings for things are not clearly stated. For instance, are the imaging conditions in Fig. 1a-i,i',ii,ii' precisely the same? Even if they are, I think it is a stretch for the authors to say that "protein levels are low" from these data.
2. Fig. 1c has a legend for "ND (not determined)" but I don't see any of that.
3. The text says that "SA-Ch is retained in the spine, thanks to the SYN tag (Figure 1c,e)", but this is not obvious from Fig. 1e, where somatic and dendritic labeling is really quite robust. Certainly, it's not clear from those images that KCl or LTP increases spine enrichment.
4. The images in Fig. 2a are incredibly misleading. That suggests that all SA-Ch signal is 100% spine-localized, when that is most emphatically not the case.
5. The legend of Fig. 2g-S-Ch says that it is not linearly dependent on SEP-GluA1, whereas the main text says that it is "[modestly dependent]". The latter seems more correct.
6. Lots of values for n are omitted, for instance in Fig. 3d.
7. The color schemes of swapping green and magenta back and forth for GFP and SA-Ch,

even within the same figure panel, is incredibly confusing.

Reviewer #3 (Remarks to the Author):

The revised manuscript by Gobbo et al. has improved significantly from the original submission. It now provides a more comprehensive analysis of the expression patterns of the activity-dependent constructs and provides much stronger support for the functionality and specificity of the new tool. Importantly, the experiments are now clearer and will be easier to reproduce by other groups. I am in favor of publication in Nature Communications, pending a few minor points which I think the authors should address, as listed below:

1. Although the manuscript now contains clearer functional data (including two-photon calcium imaging), I still believe that electrophysiology would provide valuable insight, for example by characterizing the sizes and kinetics of light-evoked EPSCs. This would convince more users to apply this tool for their own experiments.
2. In Figure 4, the authors now added new data demonstrating the impact of light-activation of AS-Ch with single-photon light while recording GCaMP6 fluorescence. These experiments are now much clearer than in the previous version, and are therefore stronger in their explanatory power, particularly in demonstrating that calcium transients triggered by light stimulation are mainly the result of the recruitment of voltage-gated calcium channels. In these experiments, it would have been good if the authors provided control measurements with a non-targeted ChETA construct, to compare the efficiency of pure synaptic stimulation vs. somatic activation.
3. In response to comments regarding the expression of native AMPA and NMDA receptors, the authors conducted immunohistochemical analysis of NMDA and AMPA subunits, demonstrating no change in the ratio between these receptors with this measure. Although electrophysiological validation would provide the ultimate test for this question, I agree with the authors' claim that the targeting motif used in the construct does not seem to change the relative expression levels. This does not, however, exclude a potential change in the expression of both types. In fact, the reference provided shows that the amplitude ratio can remain the same even though the absolute number of receptors is reduced (figure 3 in the supplied reference). I agree with the authors that it is difficult to correlate size of synapse (ROI) against absolute fluorescence intensity, which again strengthens my point that the electrophysiology characterization of AMPA/NMDA amplitudes are important.
4. In supplementary figure 7, the authors analyze the expression patterns of AS-Ch in the hippocampal region. This is a crucially important experiment, as it implies that the somatic expression which appeared to be problematic in cultured neurons does not occur in vivo. The analysis seems to indicate that AS-Ch is indeed excluded from neuronal somata and mainly concentrates in the dendrites. For clarity, it would be good if the authors supply the original images for eGFP and SA-mCherry for S7a, and also specify how exactly the 24 profiles were selected for quantification. Additionally, a non-targeted ChETA expression

construct should be included in this comparison since the expression of a membrane bound protein will always appear stronger in the dendritic regions since they contain a much larger fraction of membrane to cytoplasm.

5. While the experiments indicate that this might be a useful construct, there are nonetheless problems to be dealt with and further optimization to be done. These potential drawbacks should be laid out candidly in the discussion section, to provide the reader with a realistic understanding of how the construct might be utilized, and in which cases it would not be suitable.

Text comments:

In text you use capital dF/F and in the figure lowercase df/f - please unify.

Throughout the text: change "doxycycline" to "doxycycline"

Line 52: change "maturing" to "maturing".

Legend to Fig. 5e – please check the text next to the red and blue dots, should probably be "home cage" and "context".

There is some sort of reference problem in comparison of the images from figure 5 g and supplement figure 7 e. Both figure legends referencing to the other location for the original images (figure 5i).

Please state whether spine analyses (e.g. those in Fig. 5c-e) were done by a blinded experimenter.

Note to the Reviewers

The order and numbering of some of the Figures in the revised manuscript, particularly the Supplementary Figures, have changed in order to accommodate new data as asked by the Reviewers. In our answers we will make reference to the numbering of the previous version, but we will also indicate the number of the corresponding Figure in the revised version.

Reviewers will find their original comments in blue, and our point-by-point reply in black text underneath, and reference lines in square brackets at the end of each answer.

Reviewers' comments:

Reviewer #1 (Remarks to the Author):

The authors have addressed most concerns from me and, apparently, from other reviewers. Overall the manuscript is improved significantly. I recommend accepting the manuscript.

We thank Reviewer #1 for appreciating our efforts to improve the manuscript, and we are thankful for the valuable insights, careful reading and propositive attitude. Reviewer #1 will find a slightly modified text according to the other Reviewers' indication. However, no change to the data in the previously revised text has been made; rather, further experiments have been performed in support of those data (see comments to other reviewers).

Reviewer #2 (Remarks to the Author):

Recommendation: This revision of the manuscript is much improved. It's still not 100% clear to me how useful this reagent is going to be under difficult *in vivo* conditions, but it's a reasonable advance.

We thank Reviewer #2 for the acknowledgment. We tried to make the text clearer following the Reviewers' indications and we are grateful for highlighting points of possible difficult interpretation. We also added a discussion about potential limitations of the proposed approach under *in vivo* conditions.

Major things:

1. The manuscript is still very hard to follow (but much improved). It seems like some presentations of data are not consistent. For instance, Fig. 1c-SA-KCl-SA-Ch and Fig. 1d-SA-Ch show that KCl increases the enrichment index and positive spine fraction, but Supp. Fig. 4a would seem to indicate precisely the opposite. Another example is that the images in Fig. 2d show more punctate labeling for A-Ch than for SA-Ch, and this is presumably one of the best images obtained. Another one is that Fig. 4c shows that as much SA-Ch accumulates in spines in the dark as the light.

We revised the text thoroughly to further improve clarity and to discuss/remove the apparent highlighted inconsistencies.

We do not regard Figure 1 and Supplementary Figure 4 (same in the revised version) as contradictory. The EI calculated in Figure 1c is the enrichment of the SA-Ch protein at the spine with respect to the dendrite, and it is evaluated in the Cherry fluorescence channel. Data presented in Figure S4 are the signal, evaluated in the EGFP channel, of the MS2/RNA particles. The EGFP signal only represents the EGFP-fused MS2 protein, which is mostly bound to SA-Ch RNA due to the presence of MS2 RNA sequences in its transcript (as stated in the Methods section); please note that excess MS2 protein not bound to RNA is retained in the

nucleus (Fusco et al. 2003, *Current Biology* **13**, 572, 161–167). The EGFP signal in Figure S4 is therefore a signal for the RNA molecules. Thus, the two signals are mutually independent; the same system has been used for *Arc* DTE as in Figure 1a, where we included a pictorial representation for clarity. We do not think that the emergence of a more diffuse RNA signal is contradictory to protein translation; rather, we think that the disappearance of the punctate signal seen in unstimulated conditions is reflective of the disassembling of the RNA granule, which is permissive for the access of the ribosome to the RNA. As for Figure 2d, we included another example from different neurons in Supplementary Figure 7a, to convince that it is not a question of selecting “the best images obtained”. [84-86, 727-728]

We tried to clarify the description of data in Figure 4c (now Figure 4d). The aim of the experiment is to quantify CaMKII activation in spines that were already expressing SA-Ch. Light stimulation of SA-Ch+ spines determines CaMKII phosphorylation (and, likely, accumulation) to those spines, whereas this does not happen in spines from neurons maintained in the dark. Also, due to the temporal profile of the experimental conditions (neurons were fixed 7.5 minutes after light stimulation), it is unlikely that light stimulation has a substantial effect on SA-Ch levels. We infer this from the time course of SA-Ch expression in single spine stimulation, as reported in Figure 3d. We think that after 7.5 minutes further translation may have started, but no mature protein is present yet. Following spine activation, intracellular signal transduction and ribosome scan of the RNA, a rough estimate suggests us that at least 2 minutes are necessary for a single polypeptide chain to be synthesized, basing our calculation on the length of SA-Ch ORF and the value reported for translation speed in Morisaki et al, 2016, *Science*, **352** (6292) 1425-1429. In fact, as we report in Figure 3d, translated protein is first detectable 30 minutes after stimulation, and reaches a plateau around 60 minutes, in accordance with Aakalu et al, 2011, *Neuron*, **30** (2), 489-502. [164-181]

2. All bar plots should be presented as box plots, or even better, showing all data points like in Fig. 4a (why was this the only one to do so?). It’s just too hard to tell the reliability of the statistical tests, particularly since the authors are showing standard error instead of standard deviation (the latter would be much better).

We reported single points whenever we thought that such representation would be helpful without compromising the readability of the resulting graph and, hence, of the Figure. We now report single data points explicitly in Figure 4a and 4b in the main text, and we included an additional Figure S5 where data presented in Figure 1c and 1d are represented as box plots. For all other data presented, we did not include box plot explicitly in our figures for the sake of clarity; however, we included a table in the supplementary information (Supplementary Table 1) that includes all information that would be provided in a box plot (minimum, 1st quartile (25%), median, 3rd quartile (75%) and maximum) for the data presented in the text. By doing this, we hope that our figures convey the message in a sufficiently clear and easy-to-follow way, while still providing the reader with all the information provided by box plots. [573, 717-718]

3. The *in vivo* data in Fig. 5 isn’t super-convincing. It seems like they have to go to some lengths to find differences with the spine-neighbor calculations. Again, better presentation of the bar plot data would help evaluate these things.

Spine-neighbour calculations are just part of the analysis we made to highlight differences between the two conditions. For example, the number of SA-Ch expressing spines itself is increased in CA1 and in the DG following context exploration by the animal. We think that our spine calculations provide a good example of how our approach could be used to map potentiated synapses *in vivo*, as well as novel data regarding spine distribution following a natural stimulation like the exploration of a novel environment. These data provide a demonstration of a new experimental approach for the mapping of synapses activated in response to a given stimulus.

Following the reviewer's suggestion, we tabulated all information provided by boxplot representation in the supplementary information (Supplementary Table 1).

Lesser things:

1. The imaging settings for things are not clearly stated. For instance, are the imaging conditions in Fig. 1a-i,i',ii,ii' precisely the same? Even if they are, i think it is a stretch for the authors to say that "protein levels are low" from these data.

The imaging conditions are precisely the same between the two images, and the corresponding insets. However, we modified the text following the Reviewer's suggestion. [59-64, 706-709]

2. Fig. 1c has a legend for "ND (not determined)" but i don't see any of that.

Corrected in text.

3. The text says that "SA-Ch is retained in the spine, thanks to the SYN tag (Figure 1c,e)", but this is not obvious from Fig. 1e, where somatic and dendritic labeling is really quite robust. Certainly, it's not clear from those images that KCl or LTP increases spine enrichment.

The main purpose of Figure 1e was to show the increase in number of positive spines following the reported treatments. Indeed, for the way it is calculated, it is difficult to evaluate the enrichment index for single spines from large field images, although the lookup table we employed, which is reported under each panel, was among the most helpful ones, if not the most helpful, we could employ to highlight relative differences in intensity. Anyway, we thank the Reviewer for noting, and we modified the text where the reference appeared, making explicit reference to the somatic labeling. [99,106-109]

4. The images in Fig. 2a are incredibly misleading. That suggests that all SA-Ch signal is 100% spine-localized, when that is most emphatically not the case.

Although there are other images showing that even for SA-Ch there is residual SA-Ch fluorescence in the dendrites, (for example Figure 2f), we removed the panel in Figure 2a, to avoid the possible misleading of the reader.

5. The legend of Fig. 2g-S-Ch says that it is not linearly dependent on SEP-GluA1, whereas the main text says that it is "[modestly dependent]". The latter seems more correct.

We think that both sentences correctly describe the data. As we report in the graph below on the left, S-Ch dependence on SEP-GluA1 is better described by a nonlinear (logarithmic) fit (red line) than a linear one (blue line). Any linear distribution of data would be parallel to the diagonal in Figure 2g, as both axes are in logarithmic scale, and a slope smaller (or greater) than 1 indicates non linear dependence of data. This happens for example for SA-Ch/SEP-GluA1 data, where the linear (blue line) and the non linear fit (red line) give substantially similar fits (please see the graph below on the right). In the table in the supplementary material (Supplementary Table 1) we report the slope (\pm sem) of the linear fit of the Log value of data in Figure 2g. All fitting was performed with Graphpad Prism v6.

6. Lots of values for n are omitted, for instance in Fig. 3d.

We included single traces for the acquired spines in the new Figure 3d, and included all n values in Supplementary Table 1.

7. The color schemes of swapping green and magenta back and forth for GFP and SA-Ch, even within the same figure panel, is incredibly confusing.

We thank the Reviewer for this observation and uniformed the color schemes by showing single images in grayscale, and maintained the same color schemes for merge images.

Reviewer #3 (Remarks to the Author):

The revised manuscript by Gobbo et al. has improved significantly from the original submission. It now provides a more comprehensive analysis of the expression patterns of the activity-dependent constructs and provides much stronger support for the functionality and specificity of the new tool. Importantly, the experiments are now clearer and will be easier to reproduce by other groups. I am in favor of publication in Nature Communications, pending a few minor points which I think the authors should address, as listed below:

We thank Reviewer #3 for recognizing the work done upon revision of the original manuscript. We put our best effort in trying to address all points highlighted by Reviewer #3 (as by the other reviewers) and hope that we reasonably addressed them all. We wish to genuinely thank Reviewer #3 for the constructive approach taken during the revision of our work and useful suggestions.

1. Although the manuscript now contains clearer functional data (including two-photon calcium imaging), I still believe that electrophysiology would provide valuable insight, for example by characterizing the sizes and kinetics of light-evoked EPSCs. This would convince more users to apply this tool for their own experiments.

Along the lines suggested by the Reviewer, we compared light-evoked responses of SA-Ch with spontaneous calcium $\Delta F/F$ events recorded from neurons expressing palmitoyl-Cherry (Supplementary Figure 9). Calcium $\Delta F/F$ generated by illumination of SA-Ch look smaller than the spontaneous events we identified. We think that in order to mimic EPSCs, the photocurrent amplitude can be changed by titrating the light intensity, to exactly match the size and kinetics of spontaneous transmission, one should systematically compare different ChR2 variants. Given that their cDNA sequences usually differ in few nucleotides, we expect that changing ChETA cDNA into another ChR2 variants won't affect SA-Ch properties of spine tagging and local expression, i.e. the crucial point of the method demonstrated here. Comparing different opsins to select those with improved photocurrents, more suitable for specific applications, is left however for a different work. [293-298]

2. In Figure 4, the authors now added new data demonstrating the impact of light-activation of AS-Ch with single-photon light while recording GCaMP6 fluorescence. These experiments are now much clearer than in the previous version, and are therefore stronger in their explanatory power, particularly in demonstrating that calcium transients triggered by light stimulation are mainly the result of the recruitment of voltage-gated calcium channels. In these experiments, it would have been good if the authors provided control measurements with a non-targeted ChETA construct, to compare the efficiency of pure synaptic stimulation vs. somatic activation.

We followed the Reviewer's suggestion and we performed experiments to include non-targeted ChETA-Cherry in the measurements of light-evoked calcium $\Delta F/F$ transients. We stimulated the same region by illuminating either the synapse or the nearby dendrite, while recording from the dendrite under the spine, as previously. Illuminating the spine, but not the dendrite of SA-Ch expressing neuron, can induce depolarization that causes calcium transients (again, most likely by opening VGCCs). On the other hand, stimulation of untargeted ChETA on the dendrite, has comparable efficiency to synaptic stimulation. These new data are presented in the new Figure 4b. [158-160]

3. In response to comments regarding the expression of native AMPA and NMDA receptors, the authors conducted immunohistochemical analysis of NMDA and AMPA subunits, demonstrating no change in the ratio between these receptors with this measure. Although electrophysiological validation would provide the ultimate test for this question, I agree with the authors' claim that the targeting motif used in the construct does not seem to change the relative expression levels. This does not, however, exclude a potential change in the expression of both types. In fact, the reference provided shows that the amplitude ratio can remain the same even though the absolute number of receptors is reduced (figure 3 in the supplied reference). I agree with the authors that it is difficult to correlate size of synapse (ROI) against absolute fluorescence intensity, which again strengthens my point that the electrophysiology characterization of AMPA/NMDA amplitudes are important.

We followed the indication from Reviewer #3 and we provided electrophysiological characterization of NMDA/AMPA amplitudes from CA1 neurons of mice electroporated with doxycycline-inducible SA-Ch, along with constitutively expressed mCherry for cell identification in patch-clamp experiments. As control, we recorded from CA1 cells in the non electroporated hemisphere from the same animals. We found no difference in the NMDA/AMPA amplitudes. We decided to plot NMDA/AMPA ratio, since the recorded amplitude values for the single AMPA and NMDA currents depend on the number of stimulated synapses. However, we also found no statistically significant difference in the amplitudes of either of the two receptors between Cherry+ and Cherry- neurons. We therefore conclude that SA-Ch expression does not alter synapse physiology. The new data is presented in the new Supplementary Figure 11. Please note that we changed the representation of the surface AMPAR and NMDAR ratio in Supplementary Figure 2d for consistency with electrophysiology data. Data are now plotted as sNMDAR/sAMPAR values, and the corresponding statistical test have been performed. The change has been made purely for consistency reasons and by no means alters our previous conclusions. [77-79, 160-163, 207-212]

4. In supplementary figure 7, the authors analyze the expression patterns of AS-Ch in the hippocampal region. This is a crucially important experiment, as it implies that the somatic expression which appeared to be problematic in cultured neurons does not occur in vivo. The analysis seems to indicate that AS-Ch is indeed excluded from neuronal somata and mainly concentrates in the dendrites. For clarity, it would be good if the authors supply the original images for eGFP and SA-mCherry for S7a, and also specify how exactly the 24 profiles were selected for quantification. Additionally, a non-targeted ChETA expression construct should be included in this comparison since the expression of a membrane bound protein will always appear stronger in the dendritic regions since they contain a much larger fraction of membrane to cytoplasm.

Following Reviewer #3's suggestion, we included the original image as well as the non-electroporated hemisphere for comparison in Supplementary Figure 10. Linear profiles were taken in correspondence of all identified EGFP positive neurons; however, since the thickness of the line used to draw the profile was set to 80µm, multiple neurons may have included in a single profile. This value was chosen so that most of a single neuron could be analyzed with a single profile. We included this information in the Methods section.

We agree with Reviewer #3 that a non targeted Chr2 would also give a higher signal in the dendritic layers than in the stratum pyramidale. We therefore included in the revised paper, for comparison, the profile of slices taken from Thy1:Chr2-YFP mice that express YFP-fused Chr2 in CA1 neurons, and we repeated our analysis. However, the results of the analysis in the CA1 region show that SA-Ch and Thy1:Chr2-YFP distribution differ from each other. Most importantly, YFP fluorescence was significantly different from zero also in the whole soma layer, while this was not the case for SA-Ch. It must be noted that this comparison is conservative in evaluating differences between the two constructs, as in our analysis we normalized the resulting profile to the highest value along the spatial axis. In the hippocampus of Thy1 mice (line 18) (Jackson lab, #007612) the expression of Chr2-YFP is restricted to CA1 and it is only modestly expressed from CA3 neurons, so the fluorescence intensity we measured is due to Chr2 distribution in CA1 dendrites; however, there is still a small contribution from axons from the entorhinal cortex in the stratum radiatum close to the DG, thus leading to a slight overestimation of YFP fluorescence in the dendritic layer. The new data is integrated in the new Figure S10. These new data confirm that the subcellular expression pattern observed for SA-Ch does not merely reflect the fact that the dendritic regions contain a much larger fraction of membrane to cytoplasm. [204-206, 518-525]

5. While the experiments indicate that this might be a useful construct, there are nonetheless problems to be dealt with and further optimization to be done. These potential drawbacks should be laid out candidly in the discussion section, to provide the reader with a realistic understanding of how the construct might be utilized, and in which cases it would not be suitable.

We modified the Discussion section according to the Reviewer's indications. [243-251, 293-302]

Text comments:

In text you use capital dF/F and in the figure lowercase df/f - please unify.

Corrected in the new text and figures.

Throughout the text: change "doxycycline" to "doxycycline"

Corrected in the new text.

Line 52: change “maturing” to “maturing”.

Corrected in the new text. [56]

Legend to Fig. 5e – please check the text next to the red and blue dots, should probably be “home cage” and “context”.

We apologize for the typo, we thank Reviewer #3 for noting. We corrected the error in the new version of Figure 5.

There is some sort of reference problem in comparison of the images from figure 5 g and supplement figure 7 e. Both figure legends referencing to the other location for the original images (figure 5i).

We apologize for the mismatch in referencing the correct panels in the two Figures. We checked and corrected the reciprocal references.

Please state whether spine analyses (e.g. those in Fig. 5c-e) were done by a blinded experimenter.

We integrated the information in the Methods section. Please note that the experimenter was blind to the condition (home cage/context exposed) but not to area (DG/CA1), as dendrites from granule cells and from pyramidal neurons are evidently distinguishable from each other by morphology alone. [564-565]

REVIEWERS' COMMENTS:

Reviewer #2 (Remarks to the Author):

This paper is suitable for publication now. The authors have made it better at each revision. I appreciate the new e-phys additions, as well as the updating of the display items (and putting all data in a Supp Table). They have addressed all my concerns and it looks like those of Reviewer 3 as well.

Reviewer #3 (Remarks to the Author):

The authors have addressed the majority of my comments successfully and have explained the rationale for performing alternative experiments in other cases. I think that the addition of inducible expression constructs, and the inclusion of electrophysiological recordings have strengthened the paper and I have no further comments.

In general, this does not seem to be a tool that is exactly "ready to use", but the conceptual and technical advance is substantial and would allow the authors (and others) to follow up and further optimize the system.

Reviewer #2 (Remarks to the Author):

This paper is suitable for publication now. The authors have made it better at each revision. I appreciate the new e-phys additions, as well as the updating of the display items (and putting all data in a Supp Table). They have addressed all my concerns and it looks like those of Reviewer 3 as well.

We thank Reviewer #2 for acknowledging our efforts. We wish to thank the Reviewer for prompting us to make the paper better, and for the useful indication pointing to the portions of the text or of the figures that were not sufficiently clearly presented.

Reviewer #3 (Remarks to the Author):

The authors have addressed the majority of my comments successfully and have explained the rationale for performing alternative experiments in other cases. I think that the addition of inducible expression constructs, and the inclusion of electrophysiological recordings have strengthened the paper and I have no further comments.

In general, this does not seem to be a tool that is exactly "ready to use", but the conceptual and technical advance is substantial and would allow the authors (and others) to follow up and further optimize the system.

We thank Reviewer #3 for the valuable help in providing us meaningful indications to improve our text and for recognizing our advances. We agree that testing and, possibly, adjustments would be necessary before applying our tool to a new system, but we think that the novelty of the principle we put forward justifies the publication of our results. Anyway, we followed the Reviewer's suggestion to outline possible limitations and solutions in the Discussion of the last revision step, for which we are grateful.